



# Analysis of the distributions of hourly NO₂ concentrations contributing to annual average NO₂ concentrations across the European monitoring network between 2000 and 2014

Christopher S Malley[1], Erika von Schneidemesser[2], Sarah Moller[3], Christine F. Braban[4], W. Kevin Hicks[1], Mathew R. Heal[5]

[1]Stockholm Environment Institute, Environment Department, University of York, York, UK
[2]Institute for Advanced Sustainability Studies, Potsdam, Germany
[3]National Centre for Atmospheric Science (NCAS), Department of Chemistry, University of York, York, UK
[4]NERC Centre for Ecology & Hydrology, Penicuik, UK
[5]School of Chemistry, University of Edinburgh, Edinburgh, UK

*Correspondence to*: Christopher S. Malley (chris.malley@york.ac.uk)

**Abstract.** Exposure to nitrogen dioxide ($NO_2$) is associated with negative human health effects, both for short-term 'peak' concentrations and from long-term exposure to a wider range of $NO_2$ concentrations. For the latter, the European Union has established an air quality limit value of 40 µg m$^{-3}$ as an annual average. However, factors such as proximity and strength of local emissions, atmospheric chemistry and meteorological conditions means that there is substantial variation in the hourly $NO_2$ concentrations contributing to an annual average concentration. The aim of this analysis was to quantify the nature of this variation at thousands of monitoring sites across Europe through the calculation of a standard set of 'chemical climatology' statistics. Specifically, at each monitoring site that satisfied data capture criteria for inclusion in this analysis, annual $NO_2$ concentrations, as well as the percentage contribution from each month, hour of the day, and hourly $NO_2$ concentrations divided into 5 µg m$^{-3}$ bins were calculated.

Across Europe, 2010-2014 average annual $NO_2$ concentrations ($NO_{2AA}$) exceeded the annual $NO_2$ limit value at 8% of >2500 monitoring sites. The application of this 'chemical climatology' approach showed that sites with distinct monthly, hour of day, and hourly $NO_2$ concentration bin contributions to $NO_{2AA}$ were not grouped in specific regions of Europe, and within relatively small geographic regions there were sites with similar $NO_{2AA}$, but with differences in these contributions. Specifically, at sites with highest $NO_{2AA}$, there were generally similar contributions from across the year, but there were also differences in the contribution of peak vs moderate hourly $NO_2$ concentrations to $NO_{2AA}$, and from different hours across the day. Trends between 2000 and 2014 for 259 sites indicate that, in general, the contribution to $NO_{2AA}$ from winter months has increased, as has the contribution from the rush-hour periods of the day, while the contribution from peak hourly $NO_2$ concentrations has decreased. The variety of monthly, hour of day and hourly $NO_2$ contribution bin contributions to $NO_{2AA}$, across cities, countries and regions of Europe indicate that within relatively small geographic areas different interactions between emissions, atmospheric chemistry and meteorology produce variation in $NO_{2AA}$ and the conditions that produce it. Therefore, measures implemented



to reduce $NO_{2AA}$ in one location may not be as effective in others. The development of strategies to reduce $NO_{2AA}$ for an area should consider i) the variation in monthly, hour of day and hourly $NO_2$ concentration bin contributions to $NO_{2AA}$ within that area, and ii) how specific mitigation actions will affect variability in hourly $NO_2$ concentrations.

## 1 Introduction

Nitrogen dioxide ($NO_2$) is an air pollutant that is directly emitted and together with nitric oxide (NO) constitute primary emitted nitrogen oxides (commonly termed $NO_x$). $NO_2$ is emitted from a range of sources, most importantly from road transport (with different vehicle types emitting different proportions of $NO_x$ as $NO_2$), and is also formed in the atmosphere through the reaction between primary emitted NO and ozone ($O_3$). Exposure to $NO_2$ has been linked to negative human health effects, as summarised in the World Health Organization (WHO) Review of the Evidence on the Health Aspects of Air Pollution

(REVIHAAP, 2013). Alongside their conclusion of 'consistent' epidemiological evidence for association between short-term $NO_2$ exposure and negative respiratory effects, REVIHAAP (2013) also conclude that studies that have used multi-pollutant models to separate the association between long-term $NO_2$ exposure and respiratory effects from those of other pollutants 'provide support for an effect of $NO_2$ independent from particle mass metrics', and that this evidence was 'suggestive of a causal relationship.' The United States Environmental Protection Agency (US EPA) similarly concluded that epidemiological

and experimental evidence indicates that there is 'likely to be a causal relationship between long-term $NO_2$ exposure and respiratory effects' (US EPA, 2016). The United Kingdom Committee on the Medical Effects of Air Pollution (COMEAP) concluded that 'although it is possible that, to some extent, $NO_2$ acts as a marker of the effects of other traffic-related pollutants, the epidemiological and mechanistic evidence now suggests that it would be sensible to regard $NO_2$ as causing some of the health impact found to be associated with it in epidemiological studies' (COMEAP, 2015). Recent analysis of large prospective

cohorts in the United States and Canada also indicate significant associations between long-term $NO_2$ exposure and mortality after controlling for exposure to $PM_{2.5}$ and $O_3$ (Crouse et al., 2015; Turner et al., 2016).

In the European Union (EU), the evidence of $NO_2$ health effects has led to the establishment of air quality standards for the protection of human health. Limit values for $NO_2$ are set at 200 µg m⁻³ for 1-h average concentrations (with 18 exceedances

permitted per year), and 40 µg m⁻³ for annual average concentrations (European Council Directive 2008/50/EC, 2008). These limit values are in line with the WHO air quality guidelines (WHO, 2006). Hence a common basis for assessment of $NO_2$ levels relevant to the long-term health impact is well established across EU Member States. However, despite the consistent 'impact' metric, a common approach to assess the conditions producing this impact metric is currently lacking. For the protection of human health and vegetation, understanding these conditions is important in addition to assessment of the extent

of the impact metric itself, as it informs how the magnitude of the impact metric arises, and therefore the options available for its mitigation.



Annual average $NO_2$ concentrations in a location are determined by a combination of factors, including the following.

i) the emissions environment, such as the proximity and magnitude of major sources of $NO_x$ ($NO + NO_2$) and the proportion of $NO_x$ emissions that are directly emitted as $NO_2$, (AQEG, 2007; Beevers et al., 2012; Carslaw et al., 2011; Carslaw and Beevers, 2004; Grice et al., 2009; Vestreng et al., 2009). Across Europe, the major source of $NO_x$ emissions is road transport (39% of $NO_x$ emissions in the EU28 Member States in 2014, (EEA, 2016b)), and recent analysis has suggested that road transport $NO_x$ emissions could be even higher when accounting for real-world emission factors (Anenberg et al., 2017). Other sources include energy production and distribution (20%), commercial, institutional and households (14%), and energy use in industry (13%) (EEA, 2016b).

ii) atmospheric chemistry, including the balance between secondary $NO_2$ formation through the reaction between primary emitted NO and $O_3$, and its photolysis back to NO and $O_3$ as emissions are transported further from their source (Jenkin, 2014; Jenkin and Clemitshaw, 2000; Mavroidis and Chaloulakou, 2011). Other chemical processes which determine levels of $NO_2$ at a particular location include the reaction of $NO_2$ with the hydroxyl (OH) radical to form nitric acid ($HNO_3$), which is a major chemical removal process for $NO_2$ during daytime (Jenkin and Clemitshaw, 2000), and the combination of $NO_2$ and peroxy radicals to form peroxy acetyl nitrates (PANs) which also removes $NO_2$ close to emission sources. PAN species can also be transported long distances and act as a source of $NO_2$ remote from major emission sources (Fischer et al., 2014).

iii) atmospheric dispersion between source and receptor (AQEG, 2004; Donnelly et al., 2012). Meteorological conditions determine $NO_2$ levels by controlling the extent to which emissions can disperse (AQEG, 2004). For example, lower boundary layer heights and temperature inversions during winter can prevent pollution, including $NO_2$, from ventilating from the boundary layer, leading to higher wintertime $NO_2$ concentrations in European cities. In contrast, in summer, daytime boundary layer heights are generally higher, and there is greater solar activity which reduces $NO_2$ concentrations in some locations through $NO_2$ photolysis (Henschel et al., 2016). Additionally, the built environment also determines the extent to which road transport $NO_x$ emissions (or from other sources) can escape the street canyon, or immediate environment within which they are emitted. At background monitoring sites in rural locations, longer range transport can also determine $NO_2$ concentrations (Donnelly et al., 2012).

The combination of emission, chemistry and meteorology can drive substantial variation in the distributions of hourly $NO_2$ concentrations across the year and across the day that underpin any given annual average $NO_2$ concentration. The aim of this work was to investigate the spatial and temporal variation in the patterns of hourly $NO_2$ concentrations producing annual average $NO_2$ concentrations across Europe, utilising data from thousands of European $NO_2$ monitoring sites. At each site, a standard set of statistics were calculated to quantify the magnitude of an 'impact' metric, i.e. annual average $NO_2$ concentrations, as well as the contribution to annual average $NO_2$ concentrations from each month of the year, hour of the day,





and from hourly $NO_2$ concentrations (hourly concentrations were binned to concentration ranges in this study to investigate the influence of relatively low, moderate, or high hourly $NO_2$ concentrations in determining annual $NO_2$ concentrations). This method of integrating analysis of an impact metric with the variation in atmospheric composition producing it has been outlined previously as a 'chemical climatology' framework (Malley et al., 2014a), and applied to assess the conditions producing long-

term health relevant PM impacts, and ozone health and vegetation impacts, at the two UK European Monitoring and Evaluation Programme (EMEP) monitoring sites (Malley et al., 2015, 2016). The application here of the chemical climatology approach to assessment of long-term health-relevant $NO_2$ is the first application of this methodology to a regional monitoring network encompassing thousands of monitoring sites.

Europe-wide $NO_2$ concentrations have been examined previously. For example, Cyrys et al. (2012) investigated variation in annual $NO_2$ concentrations across 36 European cities/regions to assess spatial differences in estimated $NO_2$ exposure for use in epidemiological analysis of air pollution health impacts. Henschel et al. (2016) used monitoring data to assess trends (1999-2010) in $NO_2$ concentrations in relation to the implementation of the Euro $NO_x$ emission standards in 9 cities in 7 countries. In addition, studies focussing on one city, country, or region of Europe have also been undertaken (AQEG, 2004; Braniš, 2009;

Carslaw et al., 2011; Carugno et al., 2016; Chaloulakou et al., 2008;  Clapp and Jenkin, 2001; Cuevas et al., 2014; Donnelly et al., 2012, 2011; Gilge et al., 2010; Jenkin, 2014; Lozano et al., 2009; Mavroidis and Chaloulakou, 2011). These studies differ in the methods for analysis of $NO_2$ measurements, for example Henschel et al. (2016) select one traffic and urban background site to represent all traffic and urban background conditions in that city, while Cyrys et al. (2012) identify greater variation in annual $NO_2$ concentrations within each city than between cities but did not assess variation in short-term $NO_2$

concentration contributions to annual concentrations due to the two-week resolution of the measurements. This work builds on these previous studies by presenting a framework for the common characterisation of not only the magnitude of annual average $NO_2$ across monitoring sites, but also the variation in hourly $NO_2$ concentrations that give rise to it (Figure 1), and by applying this framework across all sites with available data. This common method of data interpretation allows the relevance of the results from previous studies assessing $NO_2$ variability in one location to be evaluated for other locations. Similarly, the

identification of the variability in hourly $NO_2$ concentration contributions to annual average $NO_2$ highlights differences in the likely effectiveness of reducing annual $NO_2$ concentrations through the implementation of mitigation strategies which may differently affect hourly $NO_2$ concentrations during particular times across the year, and across the day, or which focus on infrequent peak or frequent moderate hourly $NO_2$ concentrations. Finally, the application of this chemical climatology approach to monitoring data (or modelled data) in other regions would also allow for a consistent comparison of the variability

in hourly $NO_2$ contributions to annual average $NO_2$ between regions.



## 2 Methods

Data for this analysis was obtained from the AirBase European public data repository (https://www.eea.europa.eu/data-and-maps/data/aqereporting-2). Under the EU Air Quality Directive, $NO_2$ must be measured according to the European Standard, EN 14211, 2005 (European Council Directive 2008/50/EC, 2008). This is a chemiluminescence technique involving the

reduction of $NO_2$ to NO using heated (300–350 ºC) molybdenum surfaces followed by the gas-phase reaction between NO and $O_3$. Other measurement methods, including those used in Local Authority air quality monitoring and European Monitoring and Evaluation Programme (EMEP) monitoring, include low time resolution $NO_2$ diffusion tube measurement and higher temporal resolution photolytic converter-chemiluminesence $NO_2$ measurements, respectively. Both diffusion tubes and molybdenum converter-chemiluminscence analysers have positive biases in the measurements due to non-specific $NO_2$ chemical

measurement (Villena et al., 2012). The photolytic method has fewer artefacts but at very low $NO_2$ concentration may have interferences from PAN (Reed et al., 2016). In total, data from 3162 sites monitoring between 2000 and 2014 was assessed against the data capture criteria defined for this analysis (details below). Of the sites which met the data capture criteria, 79% measured $NO_2$ by chemiluminescence analysers (with molybdenum-converters), 1% by Differential Optical Absorption Spectroscopy (DOAS), and for the remainder the measurement method was listed as unknown. Data from all available sites in

the AirBase repository were used in this analysis, as these measurements are used to assess compliance with the EU air quality limit values. The metadata associated with each monitoring site includes its location, and its classification in terms of general area representativeness  (urban, suburban or rural) and station type (traffic, industrial or background) (European Commission, 2013; JRC-AQUILA, 2013). The combination of area and station type produce the classifications used in this paper (e.g. rural background, urban traffic).

**2.1 Chemical climate statistics**

The rationale and framework of a 'chemical climatology' analysis were outlined in Malley et al. (2014a). The aim is to link a specific impact of atmospheric composition to the conditions that produce it. To this end a chemical climate consists of three elements for which standard sets of statistics are calculated for each site (Figure 1). First, an *impact* of atmospheric composition is selected as the focus of the chemical climate. Here the impact is the risk of health effects from long-term exposure to $NO_2$,

as quantified through annual average $NO_2$ concentration (henceforth abbreviated to $NO_{2AA}$), in line with the EU regulatory guidelines. The *state* of atmospheric composition describes the underlying distributions of hourly $NO_2$ concentrations that contribute to $NO_{2AA}$. For this analysis, the percent contribution to $NO_{2AA}$ from each month of the year and hour of the day was calculated. In addition, hourly $NO_2$ concentrations for each year at each site were divided into 5 µg m$^{-3}$ bins (i.e. 0-5, 5-10, 10-15… µg m$^{-3}$) and the percent contribution to $NO_{2AA}$ from each of these bins was calculated to investigate the relative

contribution of low, moderate and high $NO_2$ concentrations to $NO_{2AA}$. These state statistics help to indicate likely causal *drivers* producing the impact metric, which, for $NO_{2AA}$, include emissions, meteorology and atmospheric chemistry (see Section 4). To characterise present-day health-relevant $NO_2$ 'chemical climates' across Europe, but to average out the influence of any



inter-annual variation in annual $NO_2$ concentrations, the statistics shown in Figure 1 were averaged between 2010 and 2014. Long-term trends in the statistics were evaluated for the period 2000 and 2014, for sites with sufficient data capture during this 15-year period.

At each site $NO_{2AA}$ was calculated if there were >75% hourly observations in that year. The 2010-2014 average was calculated if more than three of the five years had valid $NO_{2AA}$ values. These sites were then grouped into 10 µg m$^{-3}$ bins of average 2010-2014 $NO_{2AA}$ concentration, in order separately to evaluate the conditions associated with very low $NO_{2AA}$ concentrations (i.e. between 0 and 10 µg m$^{-3}$), through intermediate $NO_{2AA}$ concentrations (i.e. 10-20 µg m$^{-3}$, 20-30 µg m$^{-3}$…), to the highest observed $NO_{2AA}$ concentrations (>80 µg m$^{-3}$).

The data capture for each hour of the day, and each month of the year was also calculated for each year at each site. For those sites in each 2010-2014 $NO_{2AA}$ bins with >75% data capture in each month and each hour, the contributions of each month, hour of day, and hourly $NO_2$ concentrations in 5 µg m$^{-3}$ bins to $NO_{2AA}$ were calculated. These sites were further analysed to identify the major differences in the 'state' statistics across sites with similar 2010-2014 $NO_{2AA}$ values. The relatively low

number of sites in the 70-80 µg m$^{-3}$ and >80 µg m$^{-3}$ bins meant that variation in these statistics was examined across the sites individually. However, for the lower $NO_{2AA}$ concentration bins, which contained up to several hundred sites, Ward's hierarchical cluster analysis was used to group sites according to similarity in the contribution of different months, hours, and hourly $NO_2$ concentration bins to $NO_{2AA}$. In this process, each monitoring site initially constituted its own cluster. Clusters were then merged based on the similarity of the 2010-2014 average values of the percent contribution to $NO_{2AA}$ from January-

December, from 00:00 to 23:00, and from each 5 µg m$^{-3}$ hourly $NO_2$ concentration bin. At each step, the two clusters were merged which results in the smallest increase in within-cluster variance, and this was repeated until all sites are contained within a single cluster (Kaufman and Rousseeuw, 1990; Ward, 1963). This process is represented by a dendrogram, which was then 'cut' at an appropriate level to determine the number of clusters, and the grouping of sites. The cut-off point was determined from the visual identification of 'elbows' in the plots of proportion of variance explained against the number of

clusters into which sites were divided. The aim was to maximise the explained variance between sites with as few clusters as possible, and therefore, for each 10 µg m$^{-3}$ 2010-2014 $NO_{2AA}$ bin, the point at which increasing the number of clusters resulted in relatively small increases in the proportion of variance explained was identified. The cut-off points selected grouped sites into between 5 and 8 clusters, which explained between 57 and 75% of total between-site variance, as shown in Figures S1-S7.

Finally, sites in each of these groups which had long-running $NO_2$ measurements between 2000 and 2014 were identified. Across the 15-year period, a valid statistic (i.e. that met the data capture criteria outlined above) was required in at least 12 of the 15 years for the trend in that statistic to be estimated. The magnitude, direction and significance of trends in $NO_{2AA}$, and in month, hour of day and hourly concentration contributions to $NO_{2AA}$ were calculated using two methods that accounted for



autocorrelation in each statistic, but differed in assumptions about the type of underlying distribution of each statistic at each site. A number of parametric and non-parametric trend analysis methods were used to assess consistency of the results when different methods typically found in the literature for environmental trend analysis were applied.

Firstly, the magnitude and direction of the trend was estimated using the non-parametric Theil-Sen statistic, i.e. the median of the trends between all pairs of data points (Theil, 1950a, 1950b, 1950c). The significance of the trend was assessed using block bootstrap resampling to account for autocorrelation in each statistic at each site, using algorithms implemented in the Openair package within the R statistical software (Carslaw and Ropkins, 2012; R Core Team, 2016). The block length used was $n^{1/3}$ ($n$ = length of time series), the default in the Openair package based on the analysis in Kunsch et al. (1989). Secondly, a parametric

first-order auto-regressive (AR(1)) model was fitted to each statistic at each site, which also accounts for autocorrelation in the data (Box and Jenkins, 1976).

Trend estimations that do not account for autocorrelation in the data were also used. The non-parametric Mann-Kendall statistic was used to estimate the significance of the trend in each statistic at each site (Mann, 1945), in combination with the Theil-

Sen statistic for the magnitude and direction of the trend. Finally, trends were also estimated using parametric ordinary least-squares linear regression. These two methods did not account for autocorrelation in the trend estimation, and were therefore used to compare the consistency of results obtained with the non-parametric (Theil Sen/black bootstrap) and parametric (AR(1)) methods that did account for auto-correlation. All methods used in this work only estimated linear trends, and trends with $p \leq 0.05$ were designated as statistically significant.

## 3 Results

### 3.1 Impact metric: Annual average NO₂ concentration

Across Europe, 2587 sites met the data availability criteria to determine the 2010-2014 $NO_{2AA}$ concentration (Table 1), of which 8% exceeded the 40 µg m⁻³ EU annual $NO_2$ limit value (Figure 2). These sites were spread across Europe with 22 out of the 34 countries represented having at least one site where 2010-2014 $NO_{2AA}$ exceeded 40 µg m⁻³. Most of these exceedances

were at urban traffic sites (77%), consistent with previous analysis (EEA, 2016a). The 2010-2014 $NO_{2AA}$ concentration exceeded 40 µg m⁻³ at over 50% of urban traffic sites in Belgium, Germany, Italy, Norway, Serbia, Switzerland, and at over 40% of such sites in France, Greece, Portugal and the UK. There were also 20 urban background sites spread across cities in the UK (London, Manchester), Germany (Frankfurt), France (Paris), Spain (Madrid, Barcelona) and Italy (Rome, Monza) where 2010-2014 $NO_{2AA}$ was > 40 µg m⁻³. However, most urban background sites had 2010-2014 $NO_{2AA}$ concentrations

between 10-20 (34% of all urban background sites), 20-30 (44%) and 30-40 µg m⁻³ (18%). In contrast, most sites with the lowest concentrations were rural background sites; 61% of sites with 2010-2014 $NO_{2AA}$ concentrations in the 0-10 µg m⁻³ bin were rural background, and 93% of all rural background sites had $NO_{2AA}$ concentrations < 20 µg m⁻³.



### 3.2 State: Monthly, hour of day and hourly concentration bin contribution to annual average NO₂

Of the 2587 sites that had sufficient data capture to calculate 2010-2014 $NO_{2AA}$ concentrations, 2198 (85%) had sufficient data capture (>75%) in each month of the year and each hour of the day to assess hourly $NO_2$ contributions to $NO_{2AA}$ (Table 2). This included seven of the eight sites with 2010-2014 $NO_{2AA}$ concentrations > 80 µg m$^{-3}$, and five of the seven sites with

$NO_{2AA}$ between 70-80 µg m$^{-3}$. As described in Section 2, the large number of sites with 2010-2014 $NO_{2AA}$ in each of the lower concentration bins were clustered according to similarity in monthly, hour of day, and hourly $NO_2$ concentration bin contributions to $NO_{2AA}$ (summarised in Table 2). The proportion of sites located in different regions of Europe assigned to each cluster is shown in Figure 3 (regions are defined in Figure S8), and the locations of sites in each cluster for each $NO_{2AA}$ bin are shown in Figures S9-S15. For the majority of clusters across the 2010-2014 $NO_{2AA}$ bins, the sites were geographically

widespread across Europe. This indicates that the distinct variations in monthly, hour of day, and hourly $NO_2$ concentration bin contributions to $NO_{2AA}$ identified by the cluster analysis are not confined to specific European regions, e.g. for the 60-70 µg m$^{-3}$ 2010-2014 $NO_{2AA}$ bin, sites in Central Europe made up a substantial proportion of sites in four of the five clusters (Figure 3). However, there were also clusters that did not contain sites in particular Europe regions, and clusters that were dominated by sites from one region. Most notably there were clusters in each $NO_{2AA}$ bin between 20-30 µg m$^{-3}$ and 60-70 µg

m$^{-3}$ that predominantly contained sites from northern Italy and neighbouring areas. The distinct variations in the contribution of months, hour of day and hourly $NO_2$ concentration bins that lead to cluster separation have been summarised as (i) the differences in seasonal contribution to $NO_{2AA}$, as shown in Figures 4a and 4b, (ii) the contribution from the 6 hours with the largest contribution to $NO_{2AA}$, and the 6 hours with the lowest contribution to $NO_{2AA}$, as a proxy for the amplitude of diurnal variation in hour of day contributions to $NO_{2AA}$ (Figure 5), and (iii) the contribution from the five, ten, and fifteen 5 µg m$^{-3}$

hourly $NO_2$ concentration bins with the largest contribution to $NO_{2AA}$, as a proxy for the breadth of hourly $NO_2$ concentrations that make a substantial contribution to $NO_{2AA}$ (Figure 6).

The majority of sites where 2010-2014 $NO_{2AA}$ exceeded 40 µg m$^{-3}$ were between 40 and 70 µg m$^{-3}$. However, analysis of the conditions producing 2010-2014 $NO_{2AA}$ at the sites with highest 2010-2014 $NO_{2AA}$ (i.e. > 70 µg m$^{-3}$) illustrates patterns that

also occurred across the majority of sites that exceeded the EU limit value. All sites with highest 2010-2014 $NO_{2AA}$ concentration (> 80 µg m$^{-3}$) lacked seasonal variation in the contribution of hourly $NO_2$ concentrations to $NO_{2AA}$ (Figure 4a) and had largest contributions to $NO_{2AA}$ from daytime hours, with morning and evening peaks, reflecting their urban traffic classification. Whilst the 30-35% contribution to $NO_{2AA}$ from peak hours (Figure 5), defined as the six hours with largest contribution to $NO_{2AA}$, was similar across these sites, the timing of the peak periods differed, e.g. due to differences in the

magnitude of the morning and evening peaks in $NO_2$ concentration. For example, the top 6 hours occurred exclusively during the afternoon/evening at two of the highest $NO_{2AA}$ sites, 5 of the 6 peak hours occurred in the evening at two sites, 4 out of 6 peak hours occurred in the evening at one site, and two sites had an equal number of top 6 hours during the morning and evening (Figure 7).



In addition, the contribution from different hourly $NO_2$ concentration bins to $NO_{2AA}$ varied across these sites with highest $NO_{2AA}$. At the three French sites, a narrower range of 5 µg m$^{-3}$ concentration bins made substantial contributions to $NO_{2AA}$ compared to the three German and one UK sites. This is illustrated by the larger contribution of the top five contributing 5 µg m$^{-3}$ hourly $NO_2$ bins (27-34% of $NO_{2AA}$) at the three French sites, compared to the German and UK sites (21-22%) (Figure 6). Consequently, while $NO_{2AA}$ was similar across these sites, the frequency of short-term peak $NO_2$ concentrations was not. Hourly $NO_2$ concentrations above the 200 µg m$^{-3}$ hourly $NO_2$ EU limit value contributed 6% to $NO_{2AA}$ at GB0682A, and 2-4% at the three German sites, compared with only 0.3-1% at the French sites. Controlling hourly EU limit value exceedances would therefore have a greater impact on reducing $NO_{2AA}$ at the UK and German sites compared with the French sites. Conversely, reduction in the relatively moderate hourly $NO_2$ concentrations (60-120 µg m$^{-3}$) would yield greater reductions in $NO_{2AA}$ at these French sites (where contribution from 60-120 µg m$^{-3}$ to $NO_{2AA}$ varied between 60 and 68%) compared to the German and UK sites (45-50%). The patterns of variation in contribution from months, hour of day, and hourly $NO_2$ concentrations at the sites with 2010-2014 $NO_{2AA}$ between 70 and 80 µg m$^{-3}$ were broadly similar to those identified for the >80 µg m$^{-3}$ sites (Figures 4-6).

As shown in Section 3.1 the majority of sites where 2010-2014 $NO_{2AA}$ exceeded 40 µg m$^{-3}$ were between 40 and 70 µg m$^{-3}$. For sites with 2010-2014 $NO_{2AA}$ in the range 60-70 µg m$^{-3}$, most were urban traffic sites (Figure 2), and 61% were grouped into two of the five clusters identified (Cluster 2 and Cluster 3). These sites were geographically widespread (Figure 3, Figure S9), and had similar patterns of hourly $NO_2$ contribution to $NO_{2AA}$ as at sites with higher 2010-2014 $NO_{2AA}$. There was little variation in seasonal contribution to $NO_{2AA}$ (Figure 4a), a 30-35% contribution in the top 6 hours of the day to $NO_{2AA}$, and similar variation in contribution of hourly $NO_2$ concentration bins to $NO_{2AA}$. Cluster 2 sites had lower diurnal variation in the contribution to $NO_{2AA}$, as shown by a relatively lower contribution from the peak 6 hours of the day to $NO_{2AA}$ (31% compared with 34% for Cluster 3), and a higher contribution from the minimum 6 hours of the day compared with Cluster 3 (17% vs 13%). In addition, sites in Cluster 2 were determined by a relatively narrower set of hourly $NO_2$ concentrations, with the top five contributing 5 µg m$^{-3}$ hourly $NO_2$ bins contributing 34% of $NO_{2AA}$, compared with only 28% of $NO_{2AA}$ for Cluster 3 sites (Figure 6).

Other clusters contained sites with 2010-2014 $NO_{2AA}$ between 60 and 70 µg m$^{-3}$ where the conditions producing $NO_{2AA}$ were distinct to those sites with higher $NO_{2AA}$ (and sites in Clusters 2 & 3). For example, the three sites in Cluster 1 were in northern Italy and western Austria, and had substantial seasonal variation in contribution to $NO_{2AA}$ (Figure 4a), with 33% of $NO_{2AA}$ derived during winter, and only 18% during summer. There was also less diurnal variation in contribution to $NO_{2AA}$ than for sites in Cluster 2 and 3, with the peak 6 hours contributing 30%, and the minimum 6 hours 20% (Figure 5). In contrast, the distinction between the three sites in Cluster 4, and others was the larger diurnal variation in contribution to $NO_{2AA}$, with the peak 6 hours contributing 34%, and the minimum 6 hours only 11%. The Cluster 4 sites, in southern Germany and northern



Italy, also had the broadest range of concentrations contributing to $NO_{2AA}$, with only 24% of $NO_{2AA}$ accounted for by the top five 5 µg m$^{-3}$ hourly $NO_2$ bins.

Sites with $NO_{2AA}$ in the range 50-60 µg m$^{-3}$ were grouped into 8 clusters, with the majority (85%) grouped into four clusters
(Table 2). As for the 60-70 µg m$^{-3}$ sites, one cluster (Cluster 2) included sites predominantly in northern Italy (as well as Switzerland, southern France, and Hungary), and which had substantial seasonal variation in contribution to $NO_{2AA}$ (Figure 4a). In contrast the sites in the other 3 major clusters (Clusters 1, 3, and 4, containing 66% of sites) were more similar to those sites with higher $NO_{2AA}$ (>70 µg m$^{-3}$), with little seasonal variation in contribution to $NO_{2AA}$. The distinctions between sites in these three clusters were the extent of variation in diurnal contribution to $NO_{2AA}$ (Cluster 1 sites had the smallest difference in
contribution from the peak and minimum 6 hours, Cluster 3 had the largest (Figure 5)), and range of hourly $NO_2$ concentrations that contributed to $NO_{2AA}$ (Cluster 1 sites had the narrowest range, Cluster 3 sites the widest). Sites with 2010-2014 $NO_{2AA}$ between 50 and 60 µg m$^{-3}$ in each of these major clusters were quite geographically widespread, and located in cities throughout Europe.

For sites with 2010-2014 $NO_{2AA}$ between 40 and 50 µg m$^{-3}$, there was also a cluster (Cluster 2) with sites in northern Italy and Austria and which had the largest seasonal variation in contribution to $NO_{2AA}$ (Figure 4a). Sites in other clusters (Clusters 3 and 5) also had relatively large winter contribution to $NO_{2AA}$ compared with most sites with higher $NO_{2AA}$, and these sites were located across central and southern Europe, rather than confined to northern Italy (Figure 3, Figure S11). However, many sites with $NO_{2AA}$ in this range, spread across a large number of countries, also showed little seasonal variation in contribution
to $NO_{2AA}$. These sites grouped in Clusters 1, 4 and 6, and were distinguished by relatively low diurnal variation and narrower range of hourly $NO_2$ concentrations determining $NO_{2AA}$ for sites in Clusters 1 and 6 compared to sites in Cluster 4.

Although at the majority of sites the 2010-2014 $NO_{2AA}$ concentration did not exceed the 40 µg m$^{-3}$ EU limit value, negative health outcomes may be associated with lower concentrations (REVIHAAP, 2013). For $NO_{2AA}$ <40 µg m$^{-3}$, sites across all
clusters tended to have larger contributions from winter to $NO_{2AA}$ and lower contributions from summer. As expected, as $NO_{2AA}$ decreased there was also a narrower range of hourly $NO_2$ concentrations that contributed to $NO_{2AA}$, and therefore the top 5, 10 and 15 largest contributing concentration bins contributed a larger proportion of $NO_{2AA}$ than at sites with higher $NO_{2AA}$ (Figure 6). There were also clusters of sites in specific locations with distinct hourly $NO_2$ concentration contributions to $NO_{2AA}$. For example, in the 30-40 µg m$^{-3}$ and 20-30 µg m$^{-3}$ 2010-2014 $NO_{2AA}$ bins, there were clusters containing sites in
northern Italy and neighbouring areas (Cluster 1 and Cluster 6, respectively) that had the largest seasonal difference in contribution to $NO_{2AA}$ across all clusters in these 2010-2014 $NO_{2AA}$ bins. Similar to the sites with higher $NO_{2AA}$ concentrations, the clustering of sites with 2010-2014 $NO_{2AA}$ <40 µg m$^{-3}$ grouped sites from across wide geographic areas based on (i) a relatively narrow distribution of hourly $NO_2$ concentrations contributing to $NO_{2AA}$ (e.g. 30-40 µg m$^{-3}$, Cluster 2; 20-30 µg m$^{-3}$, Cluster 4, Figure 6), or (ii) a relatively large diurnal variation in contribution to $NO_{2AA}$ (e.g. 30-40 µg m$^{-3}$, Cluster 4 and



Cluster 7; 20-30 µg m$^{-3}$, Cluster 4, Figure 5). In addition, the contribution from morning and evening hours to those hours of the day with peak contribution also varied, as was the case for the sites with higher NO$_{2AA}$ (Figure 7).

### 3.3 Long-term trends

Trends between 2000 and 2014 were calculated at 259 sites. The estimated trends using parametric and non-parametric
approaches, and methods which did and did not account for autocorrelation produced generally consistent results for the changes in NO$_{2AA}$ and monthly, hour of day, and hourly NO$_2$ concentration bin contributions to NO$_{2AA}$. Figure 8 shows the proportion of sites with significant decreasing, increasing and non-significant trends in NO$_{2AA}$ between 2000 and 2014. The proportion of sites in each 2010-2014 NO$_{2AA}$ bin with significant decreasing trends was similar for the Theil-Sen/Mann-Kendall (Figure 8a), linear regression (Figure 8b) and first-order autoregressive (AR(1)) (Figure 8d) methods, and 20-30%
lower for the Theil-Sen/block bootstrap method (Figure 8c). However, the magnitude and direction of the trends in NO$_{2AA}$ across all sites were very similar for the Theil-Sen/block bootstrap or AR(1) methods (Figure S16). The spatial distribution of 2000-2014 changes in NO$_{2AA}$ is shown in Figure 9, and the proportion of sites in each 2010-2014 NO$_{2AA}$ bins with significant decreasing, increasing ($p \leq 0.05$), and non-significant trends in monthly, hour of day, and hourly NO$_2$ concentration bin contribution to NO$_{2AA}$ are shown in Figures 10, 11 and 12, respectively. These figures show the parametric AR(1)-estimated
trends because, in contrast to linear regression, autocorrelation was adjusted for using this method. The corresponding summaries of the non-parametric Theil-Sen/block bootstrap-derived trend estimates are provided in supplemental information for comparison, and show broadly similar patterns (Figures S17-S20). The statistics referred to below are from the AR(1) trend estimation.

The estimated trend in NO$_{2AA}$ was negative at the majority of sites. For sites where 2010-2014 NO$_{2AA}$ exceeded 40 µg m$^{-3}$, the trend was not statistically significant at a larger proportion of sites (54%, i.e. 14 out of 27 sites) than those with lower 2010-2014 NO$_{2AA}$ (Figure 8d). Sites with 2010-2014 NO$_{2AA}$ concentrations in the ranges 30-40, 20-30, 10-20, and 0-10 µg m$^{-3}$ had significant decreasing trends in NO$_{2AA}$ at 68%, 74%, 69% and 68% of sites, respectively (Figure 8). Although most sites had a non-significant change in monthly contribution to NO$_{2AA}$ between 2000 and 2014, there was a larger proportion of sites with
a significant decreasing trend in the contribution to NO$_{2AA}$ from summer months, in particular from June, and significant increasing contribution from winter months (Figure 10). The predominant pattern of change in hour of day contribution to NO$_{2AA}$ was an increasing contribution from 'peak' morning and evening hours, and a decreasing contribution to NO$_{2AA}$ from night-time hours, and hours in the middle of the day (Figure 11). For sites in the lowest 2010-2014 NO$_{2AA}$ category (0-10 µg m$^{-3}$), a smaller proportion of sites had significant increasing or decreasing trends in the contribution across the day compared
to sites with highest 2010-2014 NO$_{2AA}$. Finally, across all 2010-2014 NO$_{2AA}$ bins, the predominant pattern of change in the contribution of hourly NO$_2$ concentrations to NO$_{2AA}$ between 2000 and 2014 was an increasing contribution from relatively low hourly NO$_2$ concentrations, and a decreasing contribution from relatively high hourly NO$_2$ concentrations (Figure 12).



## 4 Discussion

The calculation of a standard set of statistics for thousands of monitoring sites showed that sites with similar $NO_{2AA}$ differed in the contribution from hourly $NO_2$ concentrations during different months, hours of the day, and across the hourly $NO_2$ concentration distribution. Clustering of sites based on these distinct contributions to $NO_{2AA}$ showed that sites with similar

$NO_{2AA}$ located relatively close to one another were often grouped into different clusters, and that sites relatively far from one another were often grouped within the same clusters (Figure 3). The observations that multiple distinct patterns contributed to $NO_{2AA}$ in a particular European region builds on analysis by Cyrys et al. (2012) showing that variation in $NO_{2AA}$ within a particular city was greater than variation in $NO_{2AA}$ across the 36 cities analysed. The present study shows that there can also be variation in the contributions from months, hours of the day, and hourly $NO_2$ concentration bins to $NO_{2AA}$ within relatively

small geographic areas across Europe

### 4.1 Drivers: Links to meteorology, emissions, and chemistry

The specific location of a monitoring site determines the relative contributions of meteorological conditions, atmospheric chemistry and $NO_x$ emission sources to the $NO_{2AA}$ concentration. In this work, sites with 2010-2014 $NO_{2AA}$ above 40 µg m$^{-3}$ were almost all urban traffic sites (JRC-AQUILA, 2013). At these sites, the lack of a seasonal pattern in contribution to $NO_{2AA}$

indicates that the major driver for $NO_{2AA}$ is the $NO_x$ emission sources in close proximity to the site. In addition to primary $NO_2$ emissions, the presence of $O_3$ at these urban traffic sites will also lead to secondary formation of $NO_2$ (from primary NO emissions). This reaction can proceed in a few seconds at polluted locations (AQEG, 2004), within the timescale of dispersion from source to roadside monitor, and suggests that long-range transport does not play the major role in driving exceedance of the $NO_{2AA}$ EU limit value. These conditions occur in cities across Europe. Schaefer et al. (2006) also showed in Hanover,

Germany, that pollutant concentrations closest to major roads in street canyons were substantially less correlated with boundary layer mixing height compared with urban background stations. Hence $NO_{2AA}$ at these sites may also be less influenced than sites with lower $NO_{2AA}$ by seasonal differences in boundary layer height, which are generally lower during the day in winter (Seidel et al., 2012). Reductions in hourly $NO_2$ concentrations across the whole year, e.g. through local $NO_x$ emission reductions, would therefore yield similar benefits in reducing $NO_{2AA}$ for the majority of sites exceeding the EU $NO_2$ annual

limit value, rather than just in winter months, owing to the relatively equal contribution from hourly $NO_2$ concentrations during all seasons (Figure 4a). However, the extent of the reduction from reducing emissions during peak hours of the day, and from reducing the short-term peak hourly $NO_2$ concentrations, varies between sites, which may reflect variability between sites in meteorological conditions or in traffic flows (i.e. strength of the emission source in close proximity to the monitoring sites).

Some sites, generally located in northern Italy and neighbouring areas, with 2010-2014 $NO_{2AA}$ between 40 and 70 µg m$^{-3}$ had substantially larger contributions from winter months to $NO_{2AA}$. This indicates that in this region the photochemical conversion of $NO_2$ to NO and $O_3$ in summer may be a more important factor in determining $NO_2$ concentrations at these sites than at other




sites across Europe with similar $NO_{2AA}$ concentrations. Northern Italy is heavily industrialised and relatively polluted region of Europe (Carugno et al., 2016; Cyrys et al., 2012; Hazenkamp-Von Arx et al., 2004). Large $NO_x$ and VOC emissions in this region, combined with relatively high levels of solar radiation in summer have previously been factors attributed to high levels of summertime photochemical ozone production (Masiol et al., 2017). Hence reducing hourly $NO_2$ concentrations during the

winter would be more effective in reducing $NO_{2AA}$, but reducing $NO_x$ emissions during the summer could reduce photochemical $O_3$ production in northern Italy and neighbouring areas. In addition to changes in atmospheric photochemistry between seasons, higher wintertime $NO_2$ concentrations have also been attributed to higher emissions from residential heating, and seasonal changes in meteorological conditions (Henschel et al., 2016). For example, Palarz et al. (2017) calculated the frequency of temperature inversions at 00:00, 06:00, 12:00, and 18:00 UTC across Europe during different seasons between

1981 and 2015. In winter, southern Europe, including northern Italy, had the highest frequency of temperature inversions at 00:00, 06:00, and 18:00 across mainland Europe. However, in summer, temperature inversions were less common in northern Italy at 06:00 compared to south-western Europe (Iberian peninsula), and at 18:00 compared to south-eastern Europe, i.e. in summer meteorological conditions were such that temperature inversions tended to dissipate earlier in northern Italy compared to the Iberian peninsula, and form later than in south-eastern Europe. This indicates that, compared to the rest of Europe, in

northern Italy there may be a larger difference between winter and summer in the frequency of temperature inversions and the trapping of pollution close to the surface that may contribute to the large difference in seasonal contribution to $NO_{2AA}$ at sites in this region.

There was also a greater contribution from winter compared to summer months to $NO_{2AA}$ for the majority of sites across Europe

with $NO_{2AA}$ below the EU limit. This likely reflects the greater distance of these sites from direct $NO_x$ emission sources, given that most of them were not urban traffic sites. The longer timescales of dispersion becomes more important in controlling $NO_2$ concentrations at these sites, and during summer $NO_2$ can be photolysed to NO and $O_3$ during transport from source to receptor. Pollutant concentrations at urban sites further from emissions sources were also shown to be more dependent on boundary layer height compared to traffic sites (Schäfer et al., 2006), which during the day are higher in summer compared to winter

(Seidel et al., 2012), increasing the volume into which $NO_x$ emissions can be dispersed.

The conditions producing $NO_{2AA}$ have changed since 2000, even though at some sites this has not resulted in a statistically significant change in $NO_{2AA}$. Between 2000 and 2014, $NO_x$ emissions from all sources were estimated to have decreased by 39%, and by 46% for road transport specifically (EEA, 2016b). However, Carslaw et al. (2011, 2016) showed that road

transport $NO_x$ emission decreases may not have been as large as expected from the implementation of vehicle emission standards. Carslaw et al. (2016) showed at roadside sites in London that reductions in $NO_2$ concentrations had been smaller than decreases in $NO_x$ concentrations between 1995 and 2015, and the $NO_2/NO_x$ emission ratio increased from less than 5% in 1997, to 25% in 2010, and 15% in 2014. The lower ratio in 2014 was attributed to a reduction in $NO_2/NO_x$ ratios in $NO_x$ emissions from heavy duty vehicles as a result of Euro IV and Euro V vehicle emissions standards. A similar increase in the





$NO_2/NO_x$ emission ratio was also shown across the EU by Grice et al. (2009). The lower than estimated reduction in $NO_x$ emissions from road transport, and the increase in the fraction of $NO_x$ emitted as $NO_2$, may be factors contributing to non-significant trends in $NO_{2AA}$ estimated at some sites, especially at the urban traffic sites with highest 2010-2014 $NO_{2AA}$.

The major changes in the hourly $NO_2$ contributions to $NO_{2AA}$ between 2000 and 2014 were a decreasing contribution from summer months, and an increasing contribution from winter months, an increasing contribution from hours during peak road traffic hours, and decreasing contributions from the highest hourly $NO_2$ concentrations. Henschel et al. (2016) analysed changes in seasonal $NO_2$ concentrations in 9 European cities and reported a decrease in summer $NO_2$ concentrations, and an increase in winter concentrations, which is consistent with the seasonal contribution trends calculated here. The present

analysis indicates that the factors contributing to enhanced wintertime $NO_2$ concentrations have become more important in determining $NO_{2AA}$ between 2000 and 2014 at the majority of sites. This includes larger $NO_x$ emissions from building heating during winter (Henschel et al., 2016), and meteorological conditions, with shallower daytime boundary layers (Seidel et al., 2012) and more frequent, deeper and stronger daytime temperature inversions in winter compared with summer (Palarz et al., 2017). Palarz et al. (2017) estimated that between 1981 and 2015, the frequency of night-time temperature inversions across

most of mainland Europe had not changed significantly in winter or summer, and during summer, there was little change in the depth and strength of these inversions. In contrast, during winter, significant decreasing trends in the depth and strength of temperature inversions were calculated across most of mainland Europe. Reductions in the depth into which emissions can disperse during wintertime temperature inversions may contribute to the increasing contribution of hourly $NO_2$ concentrations during winter to annual $NO_2$ concentrations. However, Palarz et al. (2017) did not estimate trends in the frequency, depth and

strength temperature inversions during the daytime, when the majority of $NO_x$ emissions occur, and did not specifically assess changes between 2000 and 2014 for which trends were calculated in this work.

The increasing contribution to $NO_{2AA}$ from rush-hour periods between 2000 and 2014 across sites with a wide range of 2010-2014 $NO_{2AA}$ underlines the importance of local road transport $NO_x$ emission sources in determining $NO_{2AA}$ at monitoring sites

across Europe. The increase in contribution to $NO_{2AA}$ from rush-hour periods may reflect smaller decreases in $NO_2$ concentrations during these hours compared to other hours, lower than expected decreases in total road transport $NO_x$ emissions during the period 2000-2014, as well as increases in the fraction of road transport $NO_x$ emissions emitted as $NO_2$ (Carslaw et al., 2016).

Finally, the 2000-2014 trends indicate that conditions producing the highest hourly $NO_2$ concentrations have decreased in frequency. This may reflect decreases in $NO_x$ emissions overall, but may also reflect changes in traffic patterns in particular cities (Baptista et al., 2014; CEBR, 2014; INRIX, 2016; Thomas, 2016), or meteorological conditions that concentrate $NO_2$ close to the monitoring sites. For example, Wagner et al. (2015) showed that in an urban street canyon there was a substantially stronger relationship between decreasing peak $NO_2$ concentrations and increasing mixing layer height than for mean $NO_2$





concentrations, for which there was no significant relationship. Therefore any increase in mixing layer height may be expected to disproportionately reduce highest hourly $NO_2$ concentrations as opposed to more moderate levels. Zhang et al. (2013) concluded that daytime (12:00 UTC) boundary layer height had increased between 1973 and 2010 based on daily radiosonde observations at 25 stations across Europe (average 73 m decade$^{-1}$), with no indication of seasonal differences.

Some studies have used satellite data to estimate trends in $NO_{2AA}$ across Europe (Blond et al., 2007; Geddes et al., 2016; Konovalov et al., 2010; Schneider et al., 2015). Across Europe, Geddes et al. (2016) estimate an average 1996-2012 trend in annual $NO_2$ of $-2.2$ % y$^{-1}$. The current analysis decomposes the seasonal, hour of day, and hourly $NO_2$ concentration changes that have produced this average trend, but also highlights variation in the magnitude, and statistical significance of $NO_{2AA}$

trends across Europe. Many sites with 2010-2014 $NO_{2AA}$ exceeding the annual EU limit value had a non-significant trend in $NO_{2AA}$, which may not be reflected in the satellite derived trend estimates due to the spatial resolution of the satellite $NO_2$ estimates (e.g. $0.1 \times 0.1$ degrees in Geddes et al. (2016)).

## 4.2 Implications for policies aimed at reducing annual $NO_2$ concentrations across Europe

Previous studies have shown that specific mitigation actions aimed at reducing $NO_2$ concentrations can have varying effects
in achieving this goal. For example, Jeanjean et al. (2017) modelled the effect of seven mitigation actions on mean $NO_2$ concentrations for a street in London, including additional trees lining the street, physical barriers between road and pedestrian areas, and paint applied to buildings. The individual implementation of these measures changed mean $NO_2$ concentrations beside roads between a 0.1% increase and a 7.4% decrease. Additionally, the effect of some mitigation measures on reducing $NO_2$ concentrations during a particular season, hour of day, or on peak vs moderate concentrations can vary. For example,
$NO_x$ emission reduction from reducing indoor building temperatures disproportionately occur in winter when heating is required (Chiesa et al., 2014).

The analysis presented here does not specifically identify the likely effect of individual mitigation actions aimed at reducing annual $NO_2$ concentrations on $NO_{2AA}$ at the different groups of sites identified. However, a greater understanding of the
variability in hourly $NO_2$ concentration driving $NO_{2AA}$ at different sites, for which a methodology and practical application is the focus of this paper, increases the ability to assess how changes in hourly $NO_2$ variation at a site would affect $NO_{2AA}$, e.g. due to a particular mitigation action. For example, total traffic flows in cities in the Netherlands (Almelo), and in Greece (Athens) are lower in July and August compared with winter months, and the distribution of peak traffic flow also differ, with these differences attributed to fewer commutes during the holiday period (Mavroidis and Ilia, 2012; Stathopoulos and Karlaftis,
2001; Thomas et al., 2008; Weijermars, 2007). Measures to reduce congestion in these cities may therefore disproportionately reduce traffic flows during the winter, when congestion is worse.





This work, focusing on linking hourly $NO_2$ variation to annual $NO_2$ concentrations, shows that there are sites across Europe with similar $NO_{2AA}$, but with distinct variation in seasonal contributions. With this knowledge, the effect of such a policy which disproportionately reduces winter $NO_x$ emissions may therefore be more effective at those sites within an $NO_{2AA}$ bin that have larger winter contributions to $NO_{2AA}$, and at those sites with relatively lower $NO_{2AA}$, as winter contributions to $NO_{2AA}$

tended to increase as $NO_{2AA}$ decreased. More generally, measures implemented to reduce $NO_{2AA}$ in one location may be less effective in others, and the development of strategies to reduce $NO_{2AA}$ in a particular area should consider the variation in monthly, hour of day and hourly $NO_2$ concentrations contributions to $NO_{2AA}$ within that area. Similarly, consideration should also be given to how different mitigation actions will affect hourly $NO_2$ concentrations across the year, hour of day and peak vs moderate hourly $NO_2$ concentrations, as this will determine the extent to which that measure will be effective at different

locations. In summary, if how $NO_{2AA}$ arises at sites across a region is known, then the ability of a monitoring network is enhanced to inform on how changes in hourly $NO_2$ concentrations, resulting from the implementation of a particular mitigation action, will affect an impact or regulatory metric such as annual $NO_2$ concentrations.

### 4.3 Uncertainties and Limitations

A key consideration is whether the sites used in this analysis are representative of the range of contributions from hourly $NO_2$

concentrations producing $NO_{2AA}$ across Europe. As outlined above, even within relatively small geographic areas $NO_{2AA}$, and contributions from months, hours and hourly $NO_2$ concentration bins to $NO_{2AA}$, can vary. Hence even though the $NO_2$ monitoring network across EU Member States is among the densest monitoring networks in the world, there may be locations where the combination of proximity to emission source, and factors affecting dispersion of $NO_x$ emissions mean that the contribution of months, hours of the day and hourly $NO_2$ concentrations is distinct from the monitored locations within that

city. Multiple studies have previously assessed monitoring site representativeness across Europe, using a wide variety of methods, including classifying sites based on population or other non-pollutant proxy variables (Joly and Peuch, 2012; Spangl et al., 2007), and classifying sites based on the similarity of air pollutant concentrations (Austin et al., 2013; Malley et al., 2014b; Tarasova et al., 2007). This analysis shows that, for $NO_2$, sites in close geographic proximity to one another can have different hourly $NO_2$ contributions to annual average $NO_2$, while there can be substantial similarity in those conditions at sites

distant from one another. This is in contrast to assessments of site representativeness for other pollutants, such as ozone, where distinct geographic groupings were identified from analysis of ozone datasets (Malley et al., 2014b; Tarasova et al., 2007). This work therefore emphasises that while there have been multiple methodologies outlined for classifying monitoring sites, one consideration is that a classification based on pollutant measurements is likely to be pollutant-specific, and therefore may differ depending on the pollutant selected to undertake the classification.

The trend analysis undertaken in this work estimated the linear trend in $NO_{2AA}$, and monthly, hour of day, and hourly $NO_2$ concentration bins contributions. Broadly consistent results were obtained using parametric and non-parametric methods, and with and without adjustment for autocorrelation. The proportion of sites with different trend assignments (i.e. significant





increasing, decreasing or non-significant) across the four trend estimates calculated for each statistic differed by at most 20-30%. This indicates that assumptions about the underlying distribution of the data, and the level of autocorrelation did not introduce bias into the trend estimates that might lead to different conclusions on the general patterns of change. However, changes in the drivers of $NO_2$ concentrations have not occurred linearly (Carslaw and Carslaw, 2007), and the trend analysis

does not account for these step changes and non-linearities. $NO_2$ concentrations have been shown to be affected by particular events; e.g. the 2008-2009 economic recession (Castellanos and Boersma, 2012; Cuevas et al., 2014), increases in urban buses and diesel vehicles (Carslaw and Carslaw, 2007), or changes in the $NO_2/NO_x$ emission ratio (Carslaw et al., 2016). Future work to investigate non-linear changes in the statistics calculated in this work could provide additional insight into how hourly $NO_2$ concentrations determining $NO_{2AA}$ change in response to the implementation of specific policies or mitigation strategies.

Additionally, the trend analysis was conducted at sites with sufficient data capture between 2000 and 2014. The 259 sites which met these criteria were not evenly distributed across Europe, but were disproportionately grouped in central Europe, specifically in Germany, where 53% of the sites were located, followed by Austria (15%) and France (14%). Hence the trend results presented here are determined predominantly by the specific changes that have occurred at sites in central Europe, rather than reflecting changes across the whole of Europe. A geographically broader comparison of the consistency of changes

in $NO_{2AA}$, and the conditions producing it across Europe, would be enhanced by the calculation of trends in the statistics derived here at sites in southern, eastern and northern Europe as the length of time series increase at sites in these regions, and by assessment of spatial autocorrelation across sites included in the trend analysis.

The aim of this analysis was to demonstrate that calculating a standard set of 'chemical climate' statistics at a large number of

monitoring sites could increase the information gained from the monitoring network. The calculation of a standard set of statistics has shown that, in addition to the variety of $NO_{2AA}$ across European sites shown previously (Cyrys et al., 2012), sites with similar $NO_{2AA}$ across Europe differ in how $NO_{2AA}$ is derived. Hence this study provides information as to how reducing $NO_2$ during specific months of the year, hours of the day, or focusing on peak vs moderate hourly $NO_2$ concentrations is likely to impact $NO_{2AA}$ at different types of sites across Europe. However, once emitted, $NO_2$ has multiple other impacts. For

example, $NO_2$ is a key precursor for the formation of surface $O_3$ (Karlsson et al., 2017; Monks et al., 2015), PAN (Fischer et al., 2014; Jenkin and Clemitshaw, 2000), secondary inorganic nitrate (Fuzzi et al., 2015; Putaud et al., 2010), and contributes to nitrogen deposition (Fowler et al., 2015; Galloway et al., 2008). To explore the conditions which result in the contribution of $NO_2$ to these other impacts, an alternative set of chemical climate statistics would be required. Future work to derive and apply these statistics to the European monitoring network data could allow for similarity or differences in the conditions

producing annual $NO_2$ compared to these other impacts, and therefore facilitate a more comprehensive assessment of the effects of air pollution mitigation strategies. Finally, the chemical climate statistics derived in this work have shown the variety of hourly $NO_2$ contributions producing $NO_{2AA}$ across Europe. Application of these statistics to $NO_2$ measurements from other regions could also facilitate comparison of the variation in monthly, hour of day, and hourly $NO_2$ distributions that contribute to $NO_{2AA}$ in other regions.





## 5 Conclusions

An annual average limit value for $NO_2$ concentrations for longer-term exposure of 40 µg m$^{-3}$ has been established by the European Union for the protection of human health. Analysis of monitoring network measurements is key to assessing compliance with this limit value across Europe. The aim of this analysis was to link the magnitude of this impact metric to the
monthly, hour of day, and hourly $NO_2$ concentrations variation that produces it. Hence these contributions were calculated for 2010-2014 average annual $NO_2$ concentrations for more than 2500 sites monitoring $NO_2$ across Europe, and their trends estimated at 259 sites between 2000 and 2014.

This study shows that, in general, sites with distinct monthly, hour of day, and hourly $NO_2$ concentration bin contributions to
annual $NO_2$ were not grouped into distinct geographic areas, and there were a range of patterns of hourly $NO_2$ concentration variability producing similar annual $NO_2$ concentrations with a particular area. This is a consequence of different interactions between emissions, atmospheric chemistry and meteorology at different types of site. Hence within a city, country, or European region, there is variation in the extent to which annual $NO_2$ concentrations could be reduced by focussing on, e.g. reducing peak $NO_2$ concentrations compared to more moderate concentrations, winter vs summer $NO_2$ concentrations, and rush hour
peaks compared to $NO_2$ levels across the day. Therefore, the assessment of actions to mitigate annual average $NO_2$ concentrations should evaluate their effect on hourly $NO_2$ concentrations across the year, across the day, and across the hourly $NO_2$ concentration distribution. Development of $NO_2$ mitigation strategies should take into account that across Europe a variety of seasonal and diurnal patterns of hourly $NO_2$ contribute to annual $NO_2$ concentrations, and therefore that specific measures implemented to reduce $NO_{2AA}$ in one location may not be as effective in others.

**Acknowledgements**

The authors are grateful to E. Weatherhead for insightful discussions regarding the trend analyses applied in this work. S Moller acknowledges the Natural Environment Research Council for funding (grant number NE/N005430/1).

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



**Table 1: The number and classification of sites for which 2010-2014 average annual NO$_2$ concentrations were calculated, separated into 10 µg m$^{-3}$ bins.**

| Concentration bin | Number of sites | Rural Background | Rural Industrial | Rural Traffic | Suburban Background | Suburban Industrial | Suburban Traffic | Urban Background | Urban Industrial | Urban Traffic | Other |
|---|---|---|---|---|---|---|---|---|---|---|---|
| **0-10 µg m$^{-3}$** | 317 | 193 | 22 | 0 | 25 | 20 | 1 | 13 | 11 | 1 | 31 |
| **10-20 µg m$^{-3}$** | 775 | 136 | 26 | 2 | 176 | 56 | 5 | 247 | 50 | 32 | 45 |
| **20-30 µg m$^{-3}$** | 752 | 23 | 10 | 2 | 130 | 41 | 21 | 316 | 36 | 126 | 47 |
| **30-40 µg m$^{-3}$** | 418 | 2 | 0 | 3 | 21 | 13 | 12 | 129 | 15 | 195 | 28 |
| **40-50 µg m$^{-3}$** | 202 | 0 | 0 | 4 | 1 | 5 | 16 | 16 | 1 | 145 | 14 |
| **50-60 µg m$^{-3}$** | 76 | 0 | 0 | 1 | 0 | 0 | 2 | 4 | 1 | 63 | 5 |
| **60-70 µg m$^{-3}$** | 32 | 0 | 0 | 0 | 0 | 0 | 2 | 0 | 0 | 27 | 3 |
| **70-80 µg m$^{-3}$** | 7 | 0 | 0 | 0 | 0 | 0 | 0 | 0 | 0 | 7 | 0 |
| **>80 µg m$^{-3}$** | 8 | 0 | 0 | 0 | 0 | 0 | 0 | 0 | 0 | 8 | 0 |
| **Total** | 2587 | 354 | 58 | 12 | 353 | 135 | 59 | 725 | 114 | 604 | 173 |



**Table 2: The number of sites in each 2010-2014 average annual NO₂ (NO2AA) concentration bin with sufficient data capture to calculate monthly, hour of day, and hourly concentration contributions to 2010-2014 NO2AA, split by the clusters demarcating distinct variation in monthly, hour of day, and hourly NO₂ concentration bin contributions to NO2AA.**

| Concentration bin | Number of clusters | Cluster 1 | Cluster 2 | Cluster 3 | Cluster 4 | Cluster 5 | Cluster 6 | Cluster 7 | Cluster 8 | Total |
|---|---|---|---|---|---|---|---|---|---|---|
| 0-10 µg m⁻³ | 5 | 110 | 38 | 75 | 9 | 31 | - | - | - | 263 |
| 10-20 µg m⁻³ | 7 | 121 | 182 | 102 | 154 | 75 | 23 | 2 | - | 659 |
| 20-30 µg m⁻³ | 7 | 104 | 127 | 164 | 44 | 147 | 63 | 3 | - | 652 |
| 30-40 µg m⁻³ | 8 | 30 | 27 | 59 | 42 | 78 | 39 | 35 | 34 | 344 |
| 40-50 µg m⁻³ | 8 | 31 | 12 | 14 | 35 | 27 | 34 | 15 | 7 | 175 |
| 50-60 µg m⁻³ | 8 | 18 | 12 | 9 | 16 | 5 | 1 | 3 | 1 | 65 |
| 60-70 µg m⁻³ | 5 | 3 | 11 | 7 | 4 | 3 | - | - | - | 28 |
| 70-80 µg m⁻³ | Sites analysed individually | - | - | - | - | - | - | - | - | 5 |
| >80 µg m⁻³ | Sites analysed individually | - | - | - | - | - | - | - | - | 7 |




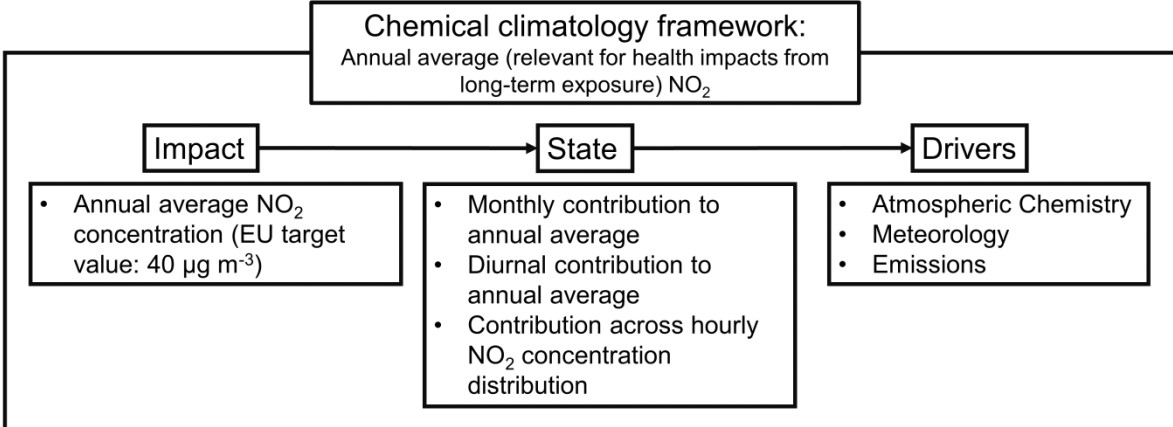

**Figure 1: Outline of Chemical Climatology elements used to evaluate the conditions producing 'long-term health-relevant' NO₂ concentrations across Europe.**





**Figure 2: Locations of NO₂ monitoring sites across Europe. Sites are grouped into 10 µg m⁻³ concentration bins based on 2010-2014 average annual NO₂ concentration, and are coloured according to the EU AirBase site-type classification.**



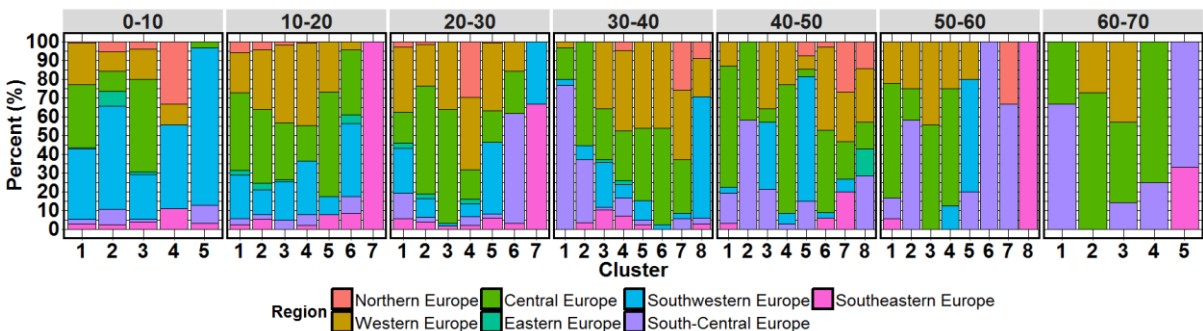

**Figure 3: The proportion of sites in each 2010-2014 average annual NO₂ (NO₂AA) 10 μg m⁻³ concentration bin in different European regions (shown in Figure S3), separated by clusters demarcating distinct variation in monthly, hour of day, and hourly NO₂ concentration bin contributions to NO₂AA.**



**Figure 4a: The contribution to 2010-2014 average annual NO₂ concentrations (NO₂ₐₐ) from spring (March, April, May), summer (June, July, August), Autumn (September, October, November), and Winter (December, January, February) months for sites split**



into bins based on the magnitude of 2010-2014 $NO_{2AA}$ concentration. For the >80 and 70-80 μg m$^{-3}$ bins, these contributions are shown for sites individually (identified by Airbase ID), while for lower concentration bins sites are grouped into clusters with distinct variation in monthly, hour of day, and hourly concentration contribution to $NO_{2AA}$. For these bins, the height of the bar shows the median contribution to 2010-2014 $NO_{2AA}$ from that season across all sites, with the 5$^{th}$ and 95$^{th}$ percentile values across the sites shows as uncertainty bars.





**Figure 4b:** The contribution to 2010-2014 average annual NO₂ concentrations (NO₂AA) from spring (March, April, May), summer (June, July, August), Autumn (September, October, November), and Winter (December, January, February) months for sites split into bins based on the magnitude of 2010-2014 NO₂AA concentration. Sites are grouped into clusters with distinct variation in monthly, hour of day, and hourly concentration contribution to NO₂AA. For these bins, the height of the bar shows the median contribution to 2010-2014 NO₂AA from that season across all sites, with the 5ᵗʰ and 95ᵗʰ percentile values across the sites shows as uncertainty bars.





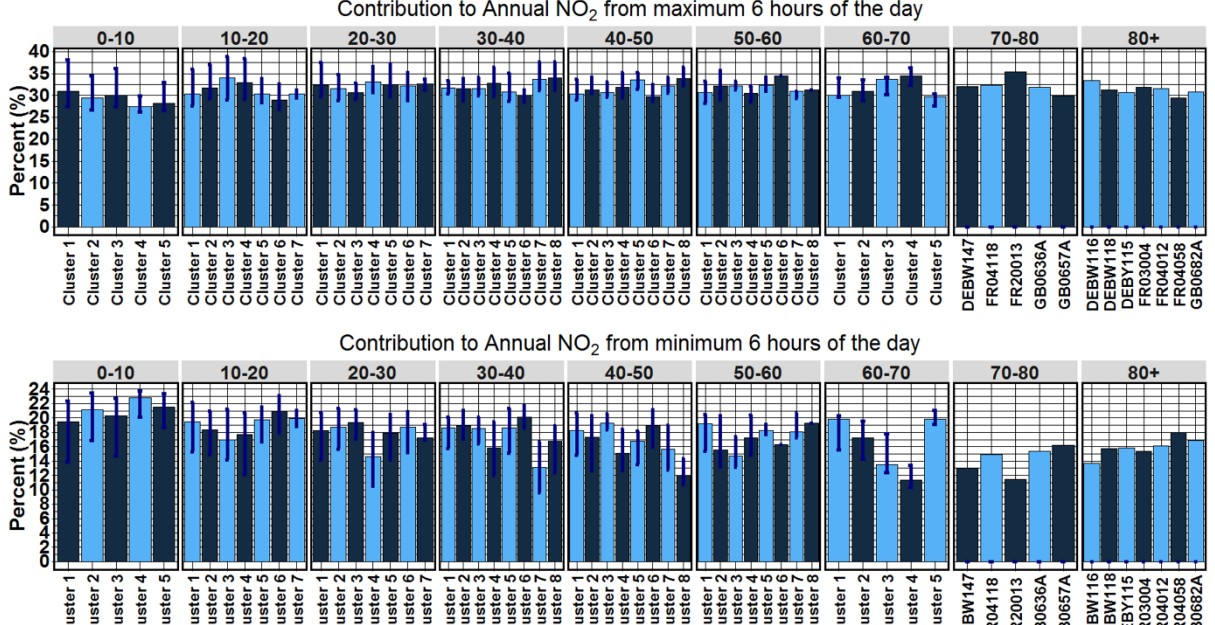

**Figure 5: The contribution to 2010-2014 average annual NO₂ concentrations (NO₂ₐₐ) from (a) the six hours with the largest contribution to 2010-2014 NO₂ₐₐ, and (b) the six hours with the smallest contribution to 2010-2014 NO₂ₐₐ, for sites separated by 2010-2014 NO₂ₐₐ concentrations, and clusters demarcating distinct variation in monthly, hour of day, and hourly NO₂ concentration bin contributions to NO₂ₐₐ. These statistics are a proxy for variation in contribution to NO₂ₐₐ across the diurnal cycle.**



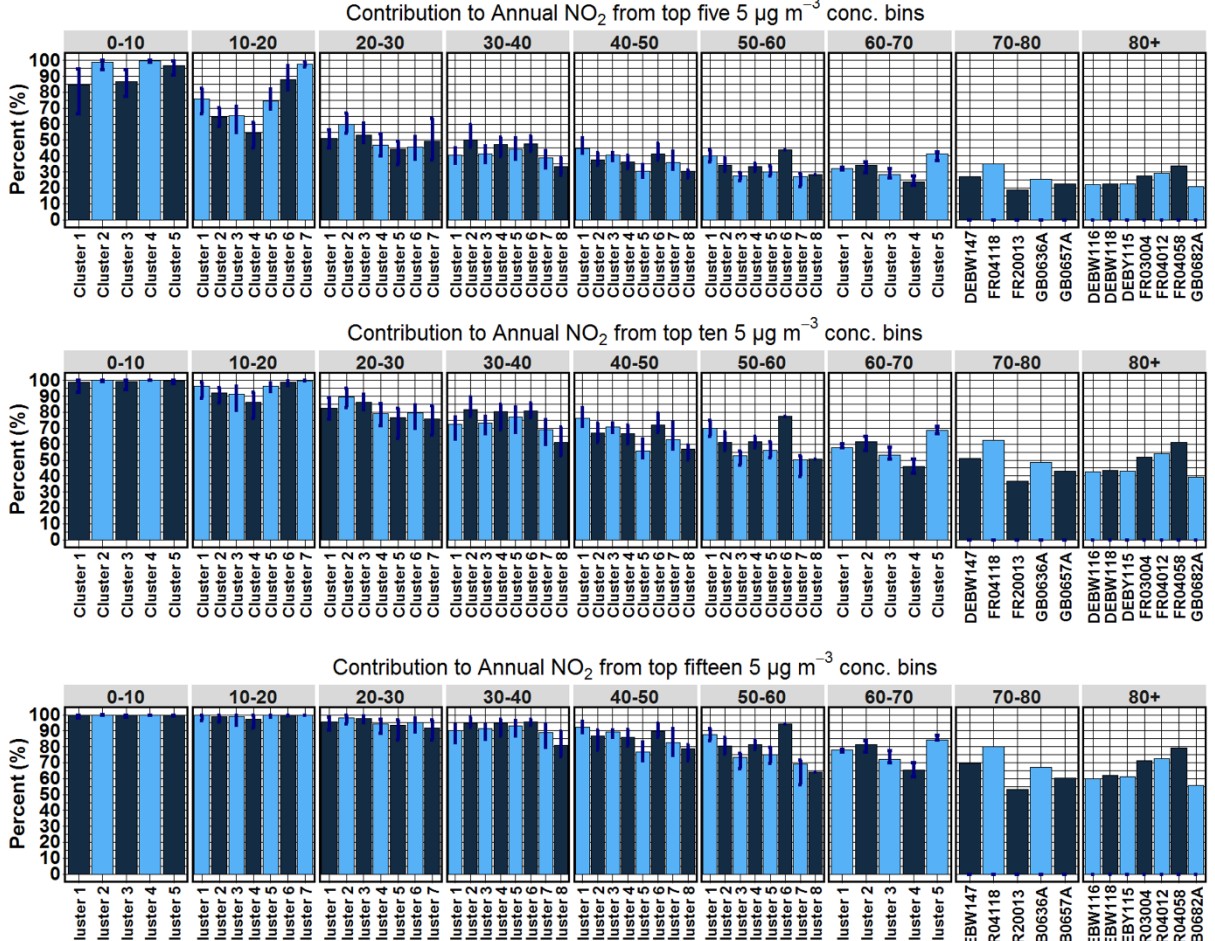

**Figure 6: The contribution to 2010-2014 average annual NO₂ concentrations (NO₂ₐₐ) from the (a) five, (b) ten, and (c) fifteen 5 μg m⁻³ hourly NO₂ concentration bins with the largest contribution to 2010-2014 NO₂ₐₐ, for sites with 2010-2014 NO₂ₐₐ in 10 μg m⁻³ bins, and separated by clusters demarcating distinct variation in monthly, hour of day, and hourly NO₂ concentration bin contributions to NO₂ₐₐ. These values are a proxy to compare the range of hourly NO₂ concentrations contributing to NO₂ₐₐ, and the near 100% contribution for the lower 2010-2014 NO₂ₐₐ bins results from the fact that for NO₂ₐₐ of 0-10 μg m⁻³, there are far fewer high hourly NO₂ concentration values.**





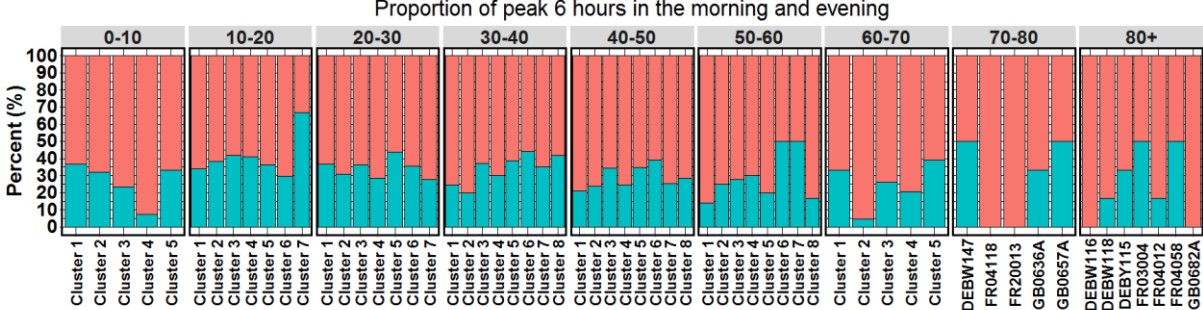

**Figure 7: Proportion of the 6 hours of the day that make the largest contribution to 2010-2014 average annual NO₂ concentrations (NO₂ₐₐ) that occurred in the morning (before 12:00, blue), and in the afternoon/evening (after 12:00, red), for sites with 2010-2014 NO₂ₐₐ in 10 μg m⁻³ bins, and separated by clusters demarcating distinct variation in monthly, hour of day, and hourly NO₂ concentration bin contributions to NO₂ₐₐ.**





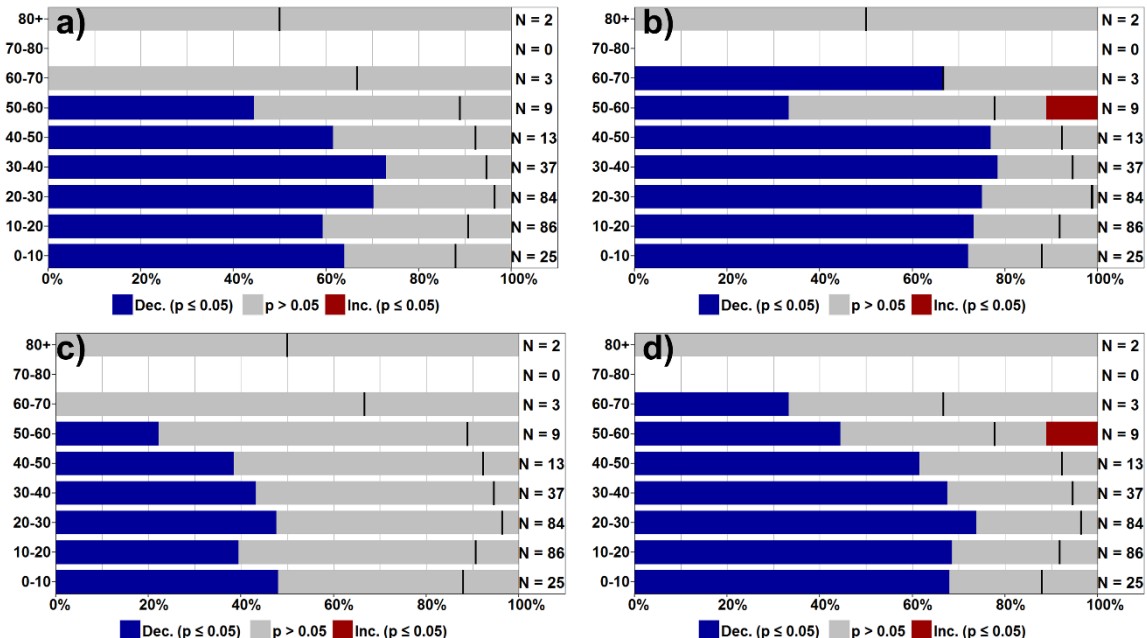

**Figure 8: Proportion of monitoring sites with 2010-2014 average annual NO$_2$ concentrations (NO$_{2AA}$) in 10 µg m$^{-3}$ bins that had significant decreasing (blue), significant increasing (red) ($p \leq 0.05$), and non-significant trends (grey) in NO$_{2AA}$ between 2000 and 2014, calculating using a) non-parametric Theil-Sen statistic and Mann Kendall test, b) parametric ordinary least-squares linear regression, c) non-parametric Theil-Sen statistic and block bootstrap resampling, d) parametric first-order autoregressive (AR(1)) model. The black line represents the division between decreasing and increasing trends within the non-significant bar.**





**Figure 9: Magnitude and significance of trend in annual average NO₂ concentrations between 2000 and 2014, with sites separated into panels based on 2010-2014 average annual NO₂ concentrations (NO₂AA). The fill colour in each point denotes the magnitude and direction of the Theil-Sen trend at a site, and the outer colour denotes whether the trend was statistically significant ($p \leq 0.05$, green), or not statistically significant ($p > 0.05$, orange). Trend estimates were calculated using a first-order autoregressive (AR(1)) model.**





**Figure 10: Proportion of sites with significant decreasing (blue), increasing (red) ($p \leq 0.05$), and non-significant (grey) trends in the monthly percentage contribution to annual average $NO_2$ between 2000 and 2014, for sites with 2010-2014 average annual $NO_2$ concentrations ($NO_{2AA}$) of a) >80 µg m$^{-3}$, b) 60-70 µg m$^{-3}$, c) 50-60 µg m$^{-3}$, d) 40-50 µg m$^{-3}$, 30-40 µg m$^{-3}$, 20-30 µg m$^{-3}$, 10-20 µg m$^{-3}$,**





**0-10 µg m⁻³. Trend estimates were calculated using a first-order autoregressive (AR(1)) model. The black line represents the division between decreasing and increasing trends within the non-significant bar.**





**Figure 11: Proportion of sites with significant decreasing (blue), increasing (red) ($p \leq 0.05$), and non-significant (grey) trends in the percentage contribution of each hour of day to annual average NO$_2$ between 2000 and 2014, for sites with 2010-2014 average annual NO$_2$ concentrations (NO$_{2AA}$) of a) >80 μg m$^{-3}$, b) 60-70 μg m$^{-3}$, c) 50-60 μg m$^{-3}$, d) 40-50 μg m$^{-3}$, 30-40 μg m$^{-3}$, 20-30 μg m$^{-3}$, 10-20 μg**





**m$^{-3}$, 0-10 μg m$^{-3}$. Trend estimates were calculated using a first-order autoregressive (AR(1)) model. The black line represents the division between decreasing and increasing trends within the non-significant bar.**





**Figure 12: Proportion of sites with significant decreasing (blue), increasing (red) (*p* < 0.05), and non-significant (grey) trends in the percentage contribution from hourly NO₂ concentrations in 5 µg m⁻³ bins to annual average NO₂ between 2000 and 2014, for sites with 2010-2014 average annual NO₂ concentrations (NO₂AA) of a) >80 µg m⁻³, b) 60-70 µg m⁻³, c) 50-60 µg m⁻³, d) 40-50 µg m⁻³, 30-40**



**µg m⁻³, 20-30 µg m⁻³, 10-20 µg m⁻³, 0-10 µg m⁻³. Trend estimates were calculated using a first-order autoregressive (AR(1)) model. The black line represents the division between decreasing and increasing trends within the non-significant bar.**