# Peer review of "Analysis of the distributions of hourly NO2 concentrations contributing to annual average NO2 concentrations across the European monitoring network between 2000 and 2014"

_Atmospheric Chemistry and Physics, 2017_

## Referee Comment (RC1) · Anonymous Referee #1 · 29 Nov 2017

This was a very comprehensive review of European NO2 measurements 2010-2014. Just some minor corrections for readers not so familiar with cluster analysis and other concepts. A few general comments may help the authors to explain some of the concepts a bit more clearly:

Cluster analysis using hourly and monthly averages: Section 2.1 line 27: mention the hourly and monthly average contribution to the annual average NO2. This is how the

cluster analysis is calculated. Could you explain how sometimes an hourly contribution might be more or less important? Does it pull out the sites that have large diurnal variability and those that stay the same. What factor is fed into the cluster analysis? Difference between hourly concentration and annual average and difference between monthly and annual? Section 2.1 Line 12 – make clearer how the 5 ugm-3 bins are different to the 10 ugm-3 bins. I will come back to this in Fig 6. Section 2.1 Line 14- can you explain what "state" statistics is? Would a Figure in the appendix showing what a cluster analysis looks like, with the branches shown and the regions in S8 put in blocks along the x axis and coloured even if we cant see the individual station names be useful? Referring to Figure 1 doesn't really help the reader so much in explaining what a cluster analysis does and looks like? In Fig 4, 5 and 6 with the explanations of what type of sites generally each cluster is made of, it is not easy to work out what the difference is. Maybe you justify in the end that there is no difference between them in most of these figures, so the cluster analysis is not specific enough to understand the characteristics of any one station?

% contribution to NO2AA 2.1, p.5 Line 26: This is used throughout and is not entirely clear. % contribution to NO2AA from each month of the year and hour of the day. Somehow I am left not entirely understanding how this was done – could it be explained better? Five , ten and fifteen 5 ugm-3 hourly NO2 Please explain this on p.8 line 19 and p. 9 line24 and in Figure 6. Has this analysis added a lot to the analysis. If you keep in, please explain how this is done a bit more clearly.

Trend analysis You look at 2000-2014 trends and compare how most recent (2010-2014) levels can see different trends from the past to the present. Have you explained how inferring a trend from the past against a level at the end of the trend may not be directly correlated to a trend forwards from now? It seems like it may be more useful to look at the 2000-2004 NO2 levels and then show whether they increased or decreased. We don't want you to do that again but just justify why the end period is useful in understanding the trend up to that point

Figures 8 and 12 are very different. Are they at odds with each other? Have you compared them? Fig 12 shows a strong linkage between NO2 level and increases or decreases and Fig 8 seems to say that there are no significant trends

Specifics: 1. Intro p.4 Line 20: New sentence: This work p.4 Line 22: I would have liked more of an explanation for Fig 1 but maybe not all readers would p.6 Line 12. . . . in each of the 201-2014 NO2aa bins. . . p. 9 Line 28. . . between 60 and 70 ugm-3 (gap) 4 Discussion: p.12, line 5 and 6. Cant this be summarised into "there is a not a regional correlation in the clustering"?

Figures Fig 2 - > 80 is clearer than 80+ Fig 4 a – >80: would be nice to label the site name in the legend/ caption?! Fig 4b line 5 caption: . . .the sites shown as.. Fig 6- comments earlier Figure 9- Zoom in (as there is nothing in N Scandinavia , N Africa, Atlantic, Eastern Europe). >80 is better than 80+. Either there are too many colour bins or it is too difficult to see the difference between a green colour and blue. It looks like green is inside but there is no green in the legend. Very difficult to see beyond the orange or green outer circle Figure 10,11 and 12- a,b,c- h is a bit hidden on Figure- move above

Appendix Figs S1-S7 could be smaller and put into one figure a- g?

---

## Referee Comment (RC2) · Anonymous Referee #2 · 2 Jan 2018

General: Well-written summary of NO2 monitoring data and trends across Europe. Ready for final publication as is. A few comments:

1) In terms of the drivers, it would have been interesting to assess correlations between actual met measurements (assuming some are nearby, if not onsite) to see if more robust assessments about the contribution drivers could be determined.

2) Wonder if a companion meteorological clustering analyses would have proven valuable, in both the trends analyses and the composition/contribution analyses (i.e., during stagnation events does one see different trends, or greater contribution from rush hours)?

3) In the US, we often don't have as much confidence in the metadata associated w/ AQ monitoring sites (e.g., site environs change over time and metadata is not updated). Appears that isn't the case in Europe, but might be interesting to "doublecheck" urban/rural & traffic/background against emissions data, or satellite landuse, or population / other surrogates for emissions.

Specific:

Page 7: May want to combine the first two full paragraphs on page 7 w/ the last paragraph from page 6.

Pag 12: Am struggling to understand sentence that starts on line 31. If those sites in N. Italy have annual averages dominated by winter months, why would photochemical drivers "be a more important" factor. Is this supposed to read "less important"?

Page 13: add "a" before "heavily industrialised".

---

## Author Comment (AC1) · 23 Jan 2018

**Reviewer 1:**

*This was a very comprehensive review of European NO2 measurements 2010-2014. Just some minor corrections for readers not so familiar with cluster analysis and other concepts. A few general comments may help the authors to explain some of the concepts a bit more clearly:*

> **Response:** We thank the reviewer for their time spent reviewing the manuscript, for their positive comments on our work, and for the minor suggestions, which have improved the paper. We respond to their general and specific suggestions below.

*Cluster analysis using hourly and monthly averages: Section 2.1 line 27: mention the hourly and monthly average contribution to the annual average NO2.*

> **Response:** We agree that specifically referring to the statistics for which increasing variance was explained is useful, and have therefore made the following amendment.

> **Original Text P7 L5:** 'The aim was to maximise the explained variance between sites with as few clusters as possible, and therefore, for each 10 µg m$^{-3}$ 2010-2014 NO$_{2AA}$ bin, the point at which increasing the number of clusters resulted in relatively small increases in the proportion of variance explained was identified.'

> **Amended Text P7 L5:** 'The aim in selecting the number of clusters of sites was to maximise the explained variance in the percentage contribution of each month, hour of day, and hourly NO$_2$ concentrations in 5 µg m$^{-3}$ bins to NO$_{2AA}$ between sites with as few clusters as possible. Therefore, for each 10 µg m$^{-3}$ 2010-2014 NO$_{2AA}$ bin, the point at which increasing the number of clusters resulted in relatively small increases in the proportion of variance explained was identified.'

*This is how the cluster analysis is calculated. Could you explain how sometimes an hourly contribution might be more or less important? Does it pull out the sites that have large diurnal variability and those that stay the same.*

> **Response:** In the cluster analysis, the grouping of sites is based on the similarity across 77 variables, the percentage contribution from 12 months of the year, 24 hours of the day, and 41 x 5 µg m$^{-3}$ hourly NO$_2$ concentration bins. At the start of the clustering, all sites are contained in individual clusters (each containing 1 site). At each step, the clusters are merged that result in the smallest increase in within cluster variance, i.e. for this first step, the two sites for which the 77 variables are most similar are grouped into one cluster. Hence the clustering algorithm is identifying groups of sites for which the pattern of percent contributions to NO$_{2AA}$ across the day, month and across the hourly NO$_2$ concentration distribution are similar. As the reviewer suggests, this can result in sites with large diurnal variability being grouped in one cluster, and those with less variability grouped in another cluster (Figure 5 shows this clearly for sites with 2010-2014 NO$_{2AA}$

between 50 and 60 $\mu g \ m^{-3}$, with sites in Cluster 4 having large diurnal variation (shown by the larger difference in contribution to $NO_{2AA}$ from the top and bottom 6 hours), and Clusters 1 and 5 having a smaller difference). In addition to grouping sites with relatively large and small differences in diurnal variation, it also highlights sites with different patterns in diurnal contribution, as shown in Figure 6, which shows a differences in the proportion of hours with large contribution to $NO_{2AA}$ occurring in the morning/evening for different types of sites.

However, the key advantage of the cluster analysis undertaken in this work is the grouping of sites based on the similarity of monthly, hour of day, and hourly $NO_2$ contribution to $NO_{2AA}$ together, rather than evaluating these contributions separately. This facilitated the identification of sites where, e.g. the contribution to $NO_{2AA}$ from hours of the day was similar, but monthly contributions, or contributions from 5 $\mu g \ m^{-3}$ hourly $NO_2$ concentration bins differed. The major distinctions and similarities between clusters are highlighted in Section 3.2 (e.g. P10 L17 highlights the similarity in monthly contribution to $NO_{2AA}$ for the two major clusters for sites with 2010-2014 $NO_{2AA}$ between 60 and 70 $\mu g \ m^{-3}$, but differences in diurnal contribution, and in contribution from across the diurnal $NO_2$ concentration distribution). The identification of similarity across clusters in some contributions to $NO_{2AA}$ (e.g. across the year), but differences in others (e.g. across the day or hourly $NO_2$ concentration distribution) underpins aspects of the discussion, e.g. P13 L26: 'However, the extent of the reduction from reducing emissions during peak hours of the day, and from reducing the short-term peak hourly NO2 concentrations, varies between sites, which may reflect variability between sites in meteorological conditions or in traffic flows (i.e. strength of the emission source in close proximity to the monitoring sites).'

We feel it is important for the reader to understand the user-defined decisions made in the clustering algorithm. These user choices include the clustering algorithm (hierarchical or non-hierarchical methods), the choice of distance measure (i.e. how the similarity across sites is quantified), and, for hierarchical methods, the method by which sites are grouped (Ward's method was used here, alternatives include Average, Minimum or Maximum linkage (Kaufman and Rousseeuw, 1990)). We have therefore acknowledged that there are various types of cluster algorithms, and that the clustering result may be affected by these choices. We have referred to a previous study that analysed different clustering algorithms, and found that Ward's method hierarchical cluster analysis performed effectively compared to alternative methods (Mangiameli et al., 1996). We have also added more text to highlight the key outputs from the cluster analysis, and directed the reader to where these are discussed.

[revised manuscript text omitted]

*What factor is fed into the cluster analysis? Difference between hourly concentration and annual average and difference between monthly and annual?*

**Response:** As outlined above, the 77 variables used to group sites encapsulated variation in the contribution to $NO_{2AA}$ across the year, day and hourly $NO_2$ concentration distribution. These are the percentage contribution to the annual average from hourly

NO$_2$ concentrations occurring in each month of the year, hour of the day, and in 5 µg m$^{-3}$ bins. This has been made clearer in the text (see additions in response to the comment directly above).

*Section 2.1 Line 12 – make clearer how the 5 ugm-3 bins are different to the 10 ugm-3 bins. I will come back to this in Fig 6.*

**Response:** We have amended this text in response to a comment below to make clear that the 5 µg m$^{-3}$ bins relate to *hourly* NO$_2$ concentrations (and the percent contribution to the annual average from hourly NO$_2$ concentrations in these ranges), as opposed to the 10 µg m$^{-3}$ bins, that disaggregate sites based on annual NO$_2$ concentrations.

*Section 2.1 Line 14-can you explain what "state" statistics is?*

**Response:** This reference to a state statistic has been replaced with an explicit explanation of the statistics that are being referenced here.

**Original Text P6 L16:** 'These sites were further analysed to identify the major differences in the 'state' statistics across sites with similar 2010-2014 NO2AA values.'

**Amended Text P6 L16:** 'These sites were further analysed to identify the major differences in the percentage contribution to NO$_{2AA}$ from each month of the year, hour of the day, and hourly NO$_2$ concentrations in 5 µg m$^{-3}$ bins across sites with similar 2010-2014 NO$_{2AA}$ values.'

*Would a Figure in the appendix showing what a cluster analysis looks like, with the branches shown and the regions in S8 put in blocks along the x axis and coloured even if we cant see the individual station names be useful?*

**Response:** The reviewer is correct that one of the advantages of the hierarchical clustering approaches (as opposed to non-hierarchical methods such as k-means) is that observations (in this case sites) can be summarised as a dendrogram. This allows the linkages between observations and clusters to be viewed, as well as the 'level' at which the dendrogram is 'cut' to produce the set of clusters. However, for this application there are a few factors that we think reduce the utility of this visualisation. Most importantly, as identified by the reviewer, there are too many sites to be individually identified on a normal sized figure, and hence it would not be possible to use this type of plot to view the linkage between individual sites. Secondly, a key conclusion of our paper is that sites with similar variation in contributions to NO$_{2AA}$ were largely not grouped into specific regions. Hence colouring the x-axis according to region would have multiple colours across each cluster, which repeats the information already included in Figure 3.

With the limitations of summarising the cluster results using a dendrogram, it was our intention in the original manuscript to use a series of figures to extract and describe the most important aspects of the cluster analysis. Firstly, it is key to understand the extent to which the number of clusters selected explains the variation between sites. We show this clearly in Figures S1-S7 for sites with 2010-2014 $NO_{2AA}$ in each 10 µg m-3 concentration bin (now Figure S2), with the number of cluster selected highlighted in red. Next, it is important that the reader understands the spatial distribution of sites assigned to each cluster, and this is shown in Figure 3 in the main manuscript, and then in supplemental maps in more detail. Finally, it is then important to understand how the variables used to group sites (i.e. percentage contribution $NO_{2AA}$ in each month, hour of day and hourly $NO_2$ distribution) vary between clusters, which are summarised in Figures 4-7.

We do not think that including dendrograms would increase the explanation of the clustering results that is already included in the current figures in the main manuscript and supplement. We do agree that it is important to clearly explain the concepts of cluster analysis, and its limitations. To this end we have expanded the description in the text of the cluster analysis methodology, the assumptions made, what is produced from the cluster analysis, and explanation as to what the different figures in the paper show in terms of the clustering results (see amended text in response to a previous comment (starting at P6 L16 in the main manuscript)).

*Referring to Figure 1 doesn't really help the reader so much in explaining what a cluster analysis does and looks like?*

    **Response:** The cluster analysis methodology is described on P6 (L16-P7 L23). We do not refer to Figure 1 in this description.

*In Fig 4, 5 and 6 with the explanations of what type of sites generally each cluster is made of, it is not easy to work out what the difference is. Maybe you justify in the end that there is no difference between them ins most of these figures, so the cluster analysis is not specific enough to understand the characteristics of any one station?*

    **Response:** As outlined above in response to a previous comment, the key advantage of the methodology adopted here is that the cluster analysis identified similarity across sites in terms of monthly, hour of day and hourly $NO_2$ concentration bin contributions to $NO_{2AA}$. This means that it was possible to determine some clusters of sites for which, e.g. the contribution of different months was similar, but for which the contribution from across the day and hourly $NO_2$ concentration distribution, varied. We synthesise the key differences between clusters (informed by Figures 4, 5, 6 and 7) within the results text of the main manuscript. Key results from the interpretation of these figures, and the clustering result in general are the larger monthly variation in contribution to $NO_{2AA}$ at sites in northern Italy, the similar contribution from months of the year at sites in other regions with high $NO_{2AA}$, the increasing winter contribution as $NO_{2AA}$ decreases, and the

distinction between sites with similar monthly contribution, but differences in hourly contribution (larger/smaller diurnal variation), and hourly $NO_2$ concentration distribution contribution (broader/narrower set of hourly $NO_2$ concentrations determining $NO_{2AA}$).

In terms of the relevance of these results to any one particular site, it is not the aim of this paper to exhaustively describe the differences between individual sites, but to explore the main differences across the European monitoring network. However, for sites with 2010-2014 $NO_{2AA}$ <70 µg m-3, where the cluster analysis was used to group sites, we do quantify the extent to which the main differences between groups of sites explain the variation between all sites in the analysis. Figures S1-S7 of the original submission (now Figure S2) show the proportion of the variation between all sites in a 10 µg m$^{-3}$ 2010-2014 $NO_{2AA}$ range that is explained by the number of clusters into which sites are grouped. This gives an indication of the remaining variability between sites within clusters that is not explored within this analysis. We have now added additional text acknowledging that there is variability between individual sites within clusters that is not considered in this analysis (see amended text in response to a comment above (P7 L18).

*% contribution to NO2AA 2.1, p.5 Line 26: This is used throughout and is not entirely clear. % contribution to NO2AA from each month of the year and hour of the day. Somehow I am left not entirely understanding how this was done – could it be explained better?*

> **Response:** We have added additional explanation as to how these statistics were calculated on first usage of this term (P6 L17).
>
> **Original Text P6 L12:** 'The data capture for each hour of the day, and each month of the year was also calculated for each year at each site. For those sites in each 2010-2014 NO2AA bins with >75% data capture in each month and each hour, the contributions of each month, hour of day, and hourly NO2 concentrations in 5 µg m$^{-3}$ bins to $NO_{2AA}$ were calculated.'
>
> **Amended Text P6 L12:** 'The data capture for each hour of the day, and each month of the year was also calculated for each year at each site. For those sites in each 2010-2014 $NO_{2AA}$ bins with >75% data capture in each month and each hour, the percent contribution of  hourly $NO_2$ concentrations in each of the 12 months, in each of the 24 hours of the day, and from hourly $NO_2$ concentrations across the year in 5 µg m$^{-3}$ bins (i.e. from hourly $NO_2$ concentrations between 0-5 µg m$^{-3}$, 5-10 µg m$^{-3}$ … 195-200 µg m$^{-3}$, >200 µg m$^{-3}$) to $NO_{2AA}$ were calculated.'

*Five , ten and fifteen 5 ugm-3 hourly NO2 Please explain this on p.8 line 19 and p. 9 line24 and in Figure 6. Has this analysis added a lot to the analysis. If you keep in, please explain how this is done a bit more clearly.*

**Response:** We have added additional explanation as to how these statistics were calculated, and have kept them in the analysis (see amended text in response to a comment above (P6 L19)). Their importance is in highlighting groups of sites where a relatively large, or relatively narrow range of hourly $NO_2$ concentrations contribute to annual $NO_2$. The EU, as well as promulgating an annual limit value, also has an hourly $NO_2$ limit value. In calculating the contribution from hourly $NO_2$ concentrations in 5 µg m$^{-3}$ bins, it is identified in this analysis that there is a contrast between sites with highest $NO_{2AA}$ at which reducing short-term peak concentrations in excess of the hourly EU limit value would also have an appreciable reduction in annual average concentrations, and those sites where short-term peak hourly $NO_2$ concentrations are less extreme, but where annual $NO_2$ concentrations are nevertheless substantially above the EU limit value.

*Trend analysis You look at 2000-2014 trends and compare how most recent (2010-2014) levels can see different trends from the past to the present. Have you explained how inferring a trend from the past against a level at the end of the trend may not be directly correlated to a trend forwards from now? It seems like it may be more useful to look at the 2000-2004 NO2 levels and then show whether they increased or decreased. We don't want you to do that again but just justify why the end period is useful in understanding the trend up to that point.*

**Response:** We agree that discussion of the relevance for assessing trends at sites grouped by their annual $NO_2$ concentration at the end of the period should be added. The reason for grouping site based on the $NO_{2AA}$ level in 2010-2014 is to provide a link to the previous section, which explores the contributions to 2010-2014 $NO_{2AA}$. The trend analysis aims to show whether these contributions, and resulting annual $NO_2$ concentrations have changed since 2000, or whether they have remained the same. This is important because over the 2000-2014 period, there have been substantial $NO_x$ emission reductions across the EU-28 Member States (EEA, 2016). The categorisation of sites based on 2010-2014 $NO_{2AA}$ shows whether $NO_{2AA}$ at sites that exceed the EU annual $NO_2$ limit value during the most recent period has decreased, or whether the changes in $NO_x$ emissions across the EU have not resulted in a decrease at these sites that currently exceed the EU standard.

We have added text to highlight these points, and to acknowledge, as the reviewer suggests, that by quantifying 2000-2014 trends at these sites, we are not making any prediction as to how trends are likely to progress into the future.

**Additional text P8 L4:** 'Trends between 2000 and 2014 were assessed by grouping all qualifying sites based on the 2010-2014 $NO_{2AA}$ value at each site. These groupings were used to investigate whether changes in drivers of $NO_2$ variability (e.g. European-wide $NO_x$ emission reductions) had had different effects at sites where the magnitude of annual $NO_2$ concentrations during the most recent period (2010-2014) differed.'

**Additional text P18 L7:** 'Finally, assessment of trends between 2000 and 2014 does not indicate how trends at each site are likely to progress into the future, which will be determined by future changes in NOx emissions, and meteorology.'

*Figures 8 and 12 are very different. Are they at odds with each other? Have you compared them? Fig 12 shows a strong linkage between NO2 level and increases or decreases and Fig 8 seems to say that there are no significant trends*

**Response:** Figures 8 and 12 are not at odds with one another. Figure 8 shows trends in *annual* $NO_2$ concentrations, while Figure 12 focusses on changes in the contribution of *hourly* $NO_2$ concentration to $NO_{2AA}$. Figure 8 does not show 'that there are no significant trends', it shows that at a substantial number of sites there has been a significant decreasing trend in annual $NO_2$ concentrations between 2000 and 2014. Figure 12 shows that at a substantial number of sites, the contribution to $NO_{2AA}$ from relatively high hourly $NO_2$ concentrations has decreased, with a corresponding increase in the contribution from relatively moderate and low hourly $NO_2$ concentrations. This general pattern of change in the distribution of hourly $NO_2$ concentrations at sites across Europe is consistent with decreasing trends in annual $NO_2$ concentrations.

*Specifics: 1. Intro p.4 Line 20: New sentence: This work*

**Response:** 'This work' is the start of the new sentence.

*p.4 Line 22: I would have liked more of an explanation for Fig 1 but maybe not all readers would*

**Response:** We have added at this point reference to Section 2.1, which describes Figure 1, and the statistics that underpin this analysis in more detail.

*p.6 Line 12. . . . in each of the 201-2014 NO2aa bins. . .*

**Response:** This sentence has been revised in line with the recommendation.

*p. 9 Line 28. . . between 60 and 70 ugm-3 (gap)*

**Response:** This sentence has been revised in line with the recommendation.

*4 Discussion: p.12, line 5 and 6. Cant this be summarised into "there is a not a regional correlation in the clustering"?*

**Response:** Yes, this is broadly what we are saying with these sentences. However, we have retained the original wording rather than that suggested by the reviewer because in some instances there were regional groupings (e.g. in northern Italy).

*Figures*

*Fig 2 - > 80 is clearer than 80+*

**Response:** We have revised all figures in line with this recommendation.

*Fig 4 a – >80: would be nice to label the site name in the legend/ caption?!*

**Response:** The site id for those sites in the 70-80 and >80 $\mu g\ m^{-3}$ is shown in the figure. For sites contained within the clusters, there are simply too many to list them in the paper.

*Fig 4b line 5 caption: . . .the sites shown as..*

**Response:** This caption has been revised in line with the recommendation.

*Fig 6- comments earlier*

**Response:** We have revised this figure to change 80+ to >80.

*Figure 9- Zoom in (as there is nothing in N Scandinavia , N Africa, Atlantic, Eastern Europe). >80 is better than 80+. Either there are too many colour bins or it is too difficult to see the difference between a green colour and blue. It looks like green is inside but there is no green in the legend. Very difficult to see beyond the orange or green outer circle*

**Response:** We have revised this figure to change 80+ to >80. We have zoomed in, and have changed how the significance of the trend is signified. Rather than the outer circle colour showing the significance of the trend, it is now the shape of each point (circle, $p < 0.05$, triangle, $p > 0.05$). Zooming in on the area of interest in the map, and having only one colour scale on the plot, shows more clearly the magnitude and significance of the trend at each site.

*Figure 10,11 and 12- a,b,c- h is a bit hidden on Figure move above*

**Response:** We have revised these figures in line with this recommendation.

*Appendix Figs S1-S7 could be smaller and put into one figure a- g?*

**Response:** We have combined these figures in line with this recommendation to form a new Figure S1.

---

## Author Comment (AC2) · 23 Jan 2018

**Reviewer 2**

*General: Well-written summary of NO2 monitoring data and trends across Europe. Ready for final publication as is. A few comments:*

**Response:** We thank the reviewer for the time they spent reviewing the manuscript, and are pleased that they consider the paper ready for publication. Below we respond to each of the reviewer's specific comments in turn.

*1) In terms of the drivers, it would have been interesting to assess correlations between actual met measurements (assuming some are nearby, if not onsite) to see if more robust assessments about the contribution drivers could be determined.*

**Response:** We agree, it would be an interesting analysis to assess variability in $NO_2$ concentrations in relation to variation in measurements of wind speed, wind direction, temperature and other meteorological parameters. However, a number of fundamental steps are required before the meteorological measurements collected across Europe, and the network of atmospheric composition measurements are effectively able to be used to undertake the analysis the reviewer outlines.

Specifically, at the majority of atmospheric composition monitoring sites there is not co-located measurement of meteorological parameters. This has recently been highlighted in the UK Air Quality Expert Group's recent analysis of the utility of the UK air quality compliance monitoring network, and the need for collocated met and atmospheric composition data was one of the key recommendations from this report: 'For some scientific and research applications the evidential value of compliance data would be greatly enhanced through the co-measurement and reporting of meteorological parameters' (AQEG, 2015).

In the absence of sufficient co-located meteorological measurements, an assessment such as that outlined by the reviewer would then rely on meteorological measurements made at nearby stations. While at rural locations nearby met and atmospheric composition sites may have a relatively high degree of representativeness, a more careful assessment of the representativeness of the urban meteorological station location, in relation to urban background and urban traffic monitoring sites would be required before the met data could be considered applicable to comparison with $NO_2$ measurements. Previous studies have highlighted i) the large gradients in local scale meteorological conditions within urban areas (Kanda, 2007), and ii) the importance of local-scale meteorology in determining $NO_2$ concentrations (Carslaw and Carslaw, 2007). We therefore consider the incorporation of meteorological measurement analysis a very substantial additional piece of analysis beyond the scope of this work. Within the

existing text, we have referred to previous studies that assess the relationship between $NO_2$ concentrations and meteorological conditions at different types of sites (e.g. P14 L24).

However, the reviewer raises an important point about the importance of meteorological conditions in determining $NO_2$ concentrations, and we have therefore referenced the above AQEG report in the discussion, and reiterated the conclusion of that report that widespread co-location of met data alongside atmospheric composition monitors could increase the ability to assess the drivers of $NO_2$ (and other air pollutant) variability, especially at urban locations.

**Additional Text P18 L23:** 'Co-located meteorological measurements with atmospheric composition measurements, as recommended by AQEG (2015), would also allow meteorological drivers of annual $NO_2$ concentrations (and other impacts) to be assessed in more detail.'

*2) Wonder if a companion meteorological clustering analyses would have proven valuable, in both the trends analyses and the composition/contribution analyses (i.e., during stagnation events does one see different trends, or greater contribution from rush hours)?*

**Response:** As outlined above, the availability of suitable meteorological data across Europe is not sufficient to undertake a meteorological cluster analysis that would be comparable with the cluster analysis of $NO_2$ measurement data. Within the manuscript we refer to previous work that has assessed the importance of meteorological conditions in determining $NO_2$ concentrations, which gives some insight into the reviewers question. Schafer et al. (2006) assessed the relationship between boundary layer height and $NO_2$ concentrations in Hannover, Germany, and found substantially less correlation with boundary layer height and $NO_2$ concentrations at roadside sites than at urban background sites. Hence whether stagnation events produces different trends, such as greater contribution from rush hour periods may depend on the location of the sites (next to a road or removed from a road).

*3) In the US, we often don't have as much confidence in the metadata associated w/ AQ monitoring sites (e.g., site environs change over time and metadata is not updated). Appears that isn't the case in Europe, but might be interesting to "doublecheck" urban/rural & traffic/background against emissions data, or satellite landuse, or population / other surrogates for emissions.*

**Response:** In line with the reviewer's request, we have 'double checked' the relationship between site classification and population and gridded $NO_x$ emission estimates. The population density from the GEOSTAT 2011 dataset

(http://ec.europa.eu/eurostat/web/gisco/geodata/reference-data/population-distribution-demography/geostat) in the 1 km grid in which the site was located was determined. Similarly, the total $NO_x$ emission in the 0.1° grid containing each site was determined from gridded $NO_x$ emissions developed by the European Monitoring and Evaluation Programme (EMEP) Centre for Emission Inventories and Projections (CEIP, http://www.ceip.at/new_emep-grid/01_grid_data_2014).

The resulting range of population density and $NO_x$ emissions in the vicinity of different classifications of monitoring sites clearly show distinctions between rural and urban/suburban sites (Figures 1 and 2 below). There is less difference between population density and $NO_x$ emissions between suburban and urban sites, and little distinction for background and traffic sites, although there is a distinction between rural industrial and other classifications of rural sites.

The lack of distinction across urban sites is likely due to the spatial resolution of the gridded population and $NO_x$ emission datasets that are available with full European coverage. One km grids, and 0.1° grids in particular will contain both roadside and background locations within a city.

We do not think that it is necessary, or advisable, to reclassify sites to differ from those used in the AirBase data repository. This is especially the case as these official sites classifications have been used in previous analyses (e.g. Joly and Peuch, 2012), and will likely be used in future work given that these classifications are underpinned by EU legislation (referenced on P5 L13 of the main paper). Creating a discrepancy with how sites are classified in this work and other studies would be unhelpful to those attempting to consider this work in the context of previous studies. More useful are studies which specifically aim to assess the classification of sites based on different variables and methodologies, of which there are multiple examples for Europe, which are referred in the discussion of this work, and to which we have now directed readers at the point in which site classification is discussed in the Methods.

Based on the reviewer's suggestion, we have now referenced previous studies that have looked at monitoring site classification across Europe in the Methods. We have not reproduced Figures 1 and 2 contained in this response document in the main text or supplement of the manuscript because i) the key conclusions of the analysis would not be affected by a small number of sites having characteristics of a different site classification, ii) these are only two proxy variables for site classification, and there may have been changes that have occurred in the site environment that affect the classification that are not reflected either in a change in population of $NO_x$ emissions at 0.1° grid size that would require site-by-site analyses to identify robustly, and iii) to keep focus on the key aspects of our paper, which relate to the contribution of hourly $NO_2$ variability to annual $NO_2$

concentrations across Europe. Specifically, for the $NO_x$ emission figure (Figure 2 below), we have not included it because of the relatively coarser scale of the gridded $NO_x$ emissions (0.1° grids). We want to avoid any confusion or suggestion that these gridded emissions represent the variability in $NO_x$ emissions within the immediate environment of the site. This is unlikely to be the case given the spatial heterogeneity of $NO_x$ emissions within a city at scales below 0.1°. We also note that this response document will be publicly available for those readers interested in understanding variation in these proxy variables in relation to the site classifications used in this work.

**Additional text P5 L19:** 'European monitoring site classification has been evaluated previously (Flemming et al., 2005; Joly and Peuch, 2012; Spangl et al., 2007).'

However, we have not included the $NO_x$ emission figure (Figure 2 below) in the supplement because of the coarser scale of the gridded $NO_x$ emissions (0.1° grids). We want to avoid any confusion or suggestion that these gridded emissions represent the variability in $NO_x$ emissions within the immediate environment of the site. This is unlikely to be the case given the spatial heterogeneity of $NO_x$ emissions within a city at scales below 0.1°.

[Figure]

**Figure 1:** Range (median, 25th and 75th percentiles at top and bottom of box, 2.5th and 97.5th percentile at top and bottom of whiskers) of estimated population density in 2011 in the 1km grids in which $NO_2$ monitoring sites of different classifications are located. Classification abbreviations denote the site area (R = Rural, S = Suburban, U = Urban), and site type (B = Background, I = Industrial, T = Traffic).

[Figure]

**Figure 2:** Range (median, 25th and 75th percentiles at top and bottom of box, 2.5th and 97.5th percentile at top and bottom of whiskers) of estimated $NO_x$ emissions in 2014 in the 0.1° grids in which $NO_2$ monitoring sites of different classifications are located. Classification abbreviations denote the site area (R = Rural, S = Suburban, U = Urban), and site type (B = Background, I = Industrial, T = Traffic).

*Specific:*

*Page 7: May want to combine the first two full paragraphs on page 7 w/ the last paragraph from page 6.*

**Response:** Paragraphs have been combined in line with the suggestion.

*Pag 12: Am struggling to understand sentence that starts on line 31. If those sites in N. Italy have annual averages dominated by winter months, why would photochemical drivers "be a more important" factor. Is this supposed to read "less important"?*

**Response:** The sentence does not state that photochemistry is an important driver in producing *high* $NO_2$ concentrations. Rather it is stating that photochemical conversion of $NO_2$ to NO and $O_3$ may be a more important driver of what the concentration of $NO_2$ is at sites in northern Italy compared to other regions. In the context of a low summertime contributions to annual average $NO_2$ at northern Italian sites, this means producing low summer $NO_2$ concentrations (and increasing $O_3$ concentrations). We have rephrased to make this clearer.

**Original text P13 L21:** 'This indicates that in this region the photochemical conversion of $NO_2$ to NO and $O_3$ in summer may be a more important factor in determining $NO_2$ concentrations at these sites than at other sites across Europe with similar $NO_{2AA}$ concentrations.'

**Amended text P13 L21:** 'This indicates that in this region the photochemical conversion of $NO_2$ to NO and $O_3$ in summer may be a more important factor in determining $NO_2$ concentrations (i.e. lowering $NO_2$ concentration during summer) at these sites than at other sites across Europe with similar $NO_{2AA}$ concentrations.'

*Page 13: add "a" before "heavily industrialised".*

**Response:** We have amended the sentence in line with the suggestion.

**References**

AQEG: Evidential Value of Defra Air Quality Compliance Monitoring. Air Quality Expert Group, Defra Publications, available at: http://uk-air.defra.gov.uk/assets/documents/reports/cat11/1509290925_DEF-PB14312_Evidential_value_of_Defra_air_quality_compliance_moni, 2015.

Carslaw, D. C. and Carslaw, N.: Detecting and characterising small changes in urban nitrogen

dioxide concentrations, Atmos. Environ., 41(22), 4723–4733, doi:10.1016/j.atmosenv.2007.03.034, 2007.

EEA: European Union emission inventory report 1990–2014 under the UNECE Convention on Long-range Transboundary Air Pollution (LRTAP). European Environment Agency Report No. 16/2016., 2016.

Flemming, J., Stern, R. and Yamartino, R. J.: A new air quality regime classification scheme for O-3, NO2, SO2 and PM10 observations sites, Atmos. Environ., 39(33), 6121–6129 [online] Available from: %3CGo, 2005.

Joly, M. and Peuch, V. H.: Objective classification of air quality monitoring sites over Europe, Atmos. Environ., 47, 111–123 [online] Available from: %3CGo, 2012.

Kanda, M.: Progress in Urban Meteorology :A Review, J. Meteorol. Soc. Japan, 85B, 363–383, doi:10.2151/jmsj.85B.363, 2007.

Kaufman, L. and Rousseeuw, P. J.: Finding Groups in Data: An Introduction to Cluster Analysis, Wiley, New York., 1990.

Mangiameli, P., Chen, S. K. and West, D.: A comparison of SOM neural network and hierarchical clustering methods, Eur. J. Oper. Res., 93(2), 402–417 [online] Available from: %3CGo, 1996.

Schäfer, K., Emeis, S., Hoffmann, H. and Jahn, C.: Influence of mixing layer height upon air pollution in urban and sub-urban areas, Meteorol. Zeitschrift, 15(6), 647–658, doi:10.1127/0941-2948/2006/0164, 2006.

Spangl, W., Schneider, J., Moosmann, L. and Nagi, C.: Representativeness and Classification of Air Quality Monitoring Stations, Umweltbundesamt Report. http://www.umweltbundesamt.at/fileadmin/site/publikationen/REP0121.pdf, 2007.

Ward, J.: Hierarchical Grouping to Optimize an Objective Function, J. Am. Stat. Assoc., 58, 236–244 [online] Available from: http://www.jstor.org/stable/2282967, 1963.